 SciPost Phys. Lect.Notes 39 (2022)

# Berry-Chern monopoles and spectral flows

**Pierre A. L. Delplace**

Ens de Lyon, CNRS, Laboratoire de Physique, F-69342 Lyon, France

## Abstract

This lecture note adresses the correspondence between spectral flows, often associated to unidirectional modes, and Chern numbers associated to degeneracy points. The notions of topological indices (Chern numbers, analytical indices) are introduced for non specialists with a wave physics or condensed matter background. The correspondence is detailed with several examples, including the Dirac equations in two dimensions, Weyl fermions in three dimensions, the shallow water model and other generalizations.

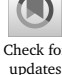

# 1 Introduction

In his successful popular book La valeur de la science [1], the mathematician physicist Henri Poincaré pointed out in 1905 that : "*we can see mathematical analogies between phenomena which have no physical relation neither apparent nor real, so that the laws of one of these phenomena help us to guess those of the other. [...] The goal of mathematical physics is not only to facilitate the physicist in numerical computation [...]. It is still, it is above all to make the physicists know the hidden harmony of things by showing them from a new angle.*" As one reads these words today, it is striking how much they make one think about the role of topology in physics.

Topology spread in virtually all branches of physics as it helps unveiling the hidden connections between apparently different phenomena. In condensed matter physics, its use has been greatly accelerated since the rise of topological insulators [2, 3], one century after Poincaré's words. In that context, topology deals with the unavoidable singular properties of electronic wavefunctions associated to the energy bands of solids, and is thus, for that reason, referred to as *band topology*. Those singularities are classified by integer numbers, called *topological invariants*, that allow physicists to classify band insulators, superconductors and semimetals, beyond the realm of phase transitions of statistical physics [4–8]. Although those topological properties are somehow *hidden*, as they appear in a quite abstract space, their physical consequences are observable and manifest for instance by a robust uni-directional transport.

One of the examples chosen by Poincaré to illustrate his statement is the wave equation, as it establishes a deep connection between sound, light and radio communications. It is thus not so surprising that the topological approaches that were developed in the realm of single particles quantum physics, which is essentially wave physics, naturally spread to various classical waves systems, in optics, acoustics, mechanics, plasmas and fluids, even at geo/astrophysical scales [9–15]. All those systems share the same *hidden harmony* (to quote Poincaré), that topology reveals by making us see those very different systems from a common angle. Topology thus basically provided a new perspective in the description of waves and a powerful tool for their manipulation.

This manuscript is dedicated to a key concept in wave topology, that is common to various quantum and classical systems. It was actually already put forward by Volovik to draw formal

analogies between $^3$He-A superfluids and the standard model in high energy physics [16]. Here we want to present it in a more wave physics / quantum condensed matter mind oriented way. We shall refer to that concept as the *monopole - spectral flow correspondence*. In short, this correspondence relates the topological properties of point defects in three dimensional (3D) space in which eigenstates of the system are parametrized, to the existence of peculiar modes that transit from an energy (or frequency) branch to another one; a phenomenon which is referred to as a *spectral flow*. Those point defects correspond to degeneracy points (or band crossing points), that act as the source of a *Berry curvature*, and whose flux through a close surface surrounding them is quantized and expressed by a topological invariant, the *first Chern number*. For that reason, we shall refer to these special points as *Berry-Chern* monopoles. The spectral flow, on the other hand, consists by construction to modes that propagate in a fixed direction. The monopole-spectral flow correspondence is a direct link between topology and unidirectional wave transport.

The goal of this manuscript is to introduce this correspondence in a pedagogical way, and make it accessible and useful to non specialists. In a first part, we focus on canonical examples encountered in condensed matter physics that we treat in a unified way. We use a minimal two-band model to illustrate this ubiquitous correspondence and describe simultaneously interface states in 2D systems such as Chern insulators, quantum valley Hall insulators and other classical waves systems, together with the dispersive Landau levels that trigger the chiral anomaly in 3D Weyl semimetals [17, 18]. This first part is mainly dedicated to a broad community of quantum and classical physicists that are not familiar with the topological concepts. Basic notions such as the Berry curvature, the first Chern number and the Weyl quantization are introduced, detailed and illustrated. The second part of this manuscript is dedicated to a far richer model, first proposed by Ezawa in the context of semimetals in a magnetic field [19], that allows for an investigation of the correspondence beyond the standard linear two-band crossing framework discussed in the first part. We discuss in details the structure of the spectral flow and its associated eigenmodes. In particular, this model is based on a spin algebra rather than a Clifford algebra, and thus goes beyond the usual framework of Dirac operators where this correspondence has been extensively studied (see e.g. [20–25] and the textbook [26]). We thus introduce an index for this "non-Dirac" operator, that accounts for the spectral flow of the model, and we compute it. We also derive a general expression for the Chern numbers, and show by an explicit calculation that it is equal to the index we have introduced. This makes clear the intervention of the Atiyah-Singer index theorem to understand topological interface modes and 3D current of Landau levels associated to anomalies beyond two-band models. The unusual Landau levels behavior of effective spin$-J$ quasiparticles protected by crystal symmetries [19, 27] and the two eastward topological equatorial oceanic waves [15] are both direct examples accounted by this model.

## 2    Spectral flows in two-band models from archetypal examples in condensed matter physics

Let us first revisit two canonical condensed matter examples where spectral flows emerge : the 2D Dirac Hamiltonian with an anisotropic mass term, and the 3D Weyl Hamiltonian under a constant axial magnetic field.

## 2.1 2D Dirac fermions with an anisotropic mass

The simplest and certainly the most ubiquitous physical model that manifests a spectral flow is that of a 2D massive Dirac Hamiltonian

$$H_D = c \begin{pmatrix} m(x)c & \hat{p}_x - i\hat{p}_y \\ \hat{p}_x + i\hat{p}_y & -m(x)c \end{pmatrix}, \tag{1}$$

where $c$ is a celerity (e.g. the Fermi velocity) and $\hat{\mathbf{p}} = (\hat{p}_x, \hat{p}_y)$ is the momentum operator. Importantly here, the mass term $m(x)$ varies in space along the $x$ direction, and is assumed to change sign at a given $x_0$ that we choose to be $x_0 = 0$. For simplicity, we shall linearize $m$ around $m(0) = 0$ so that $m(x) = \beta x$ with $\beta \equiv \partial_x m$. This is for convenience, as the properties we aim at describing do not depend on this linearization. The Hamiltonian (1) describes e.g. the band inversion that occurs at the interface between two topologically insulators when tuning a parameter that regulates the amplitude of the gap, that is the mass term [28, 29].

Then, because of the $x$ dependence of the mass, $p_x$ is not a good quantum number, unlike $p_y$. One can then rewrite the Dirac Hamiltonian in position representation for plane waves of wavenumber $k_y$ in the $y$ direction by substituting $\hat{p}_x = -i\hbar \partial_x$, as

$$H_D = c \begin{pmatrix} c\beta x & -i\hbar \partial_x - i\hbar k_y \\ -i\hbar \partial_x + i\hbar k_y & -c\beta x \end{pmatrix}. \tag{2}$$

We aim at deriving the spectrum of this Hamiltonian as a function of the wavenumber $k_y$. For that purpose, it is convenient to first apply a unitary transformation $\tilde{H}_D = R H_D R^{-1}$ so that $k_y$ becomes proportional to the diagonal Pauli matrix $\sigma_z$. Writing $H_D = -i\hbar c \partial_x \sigma_x + \hbar c k_y \sigma_y + \beta c^2 x \sigma_z$ where the $\sigma_i$'s are the Pauli matrices, such a unitary transformation $R$ amounts to a spin rotation where each axis are cyclically interchanged, and thus

$$\tilde{H}_D = \begin{pmatrix} \hbar c k_y & \beta c^2 x - \hbar c \partial_x \\ \beta c^2 x + \hbar c \partial_x & -\hbar c k_y \end{pmatrix}. \tag{3}$$

Let us then introduce the characteristic length $\ell \equiv \sqrt{\hbar/\beta c}$ that allows us to rewrite the Dirac Hamiltonian in terms of annihilation and creation operators

$$\hat{a} = \frac{1}{\sqrt{2}} \left( \frac{x}{\ell} + \ell \partial_x \right), \qquad \hat{a}^\dagger = \frac{1}{\sqrt{2}} \left( \frac{x}{\ell} - \ell \partial_x \right) \tag{4}$$

that satisfy $[a, a^\dagger] = 1$ and act on *number states* $|n\rangle$ as

$$\hat{a} |n\rangle = \sqrt{n} |n-1\rangle, \qquad \hat{a}^\dagger |n\rangle = \sqrt{n+1} |n+1\rangle, \qquad \hat{a}^\dagger \hat{a} |n\rangle = n |n\rangle. \tag{5}$$

We recall that number states are related to Hermite functions $\varphi_n(x)$ following

$$\langle x|n\rangle = \varphi_n(x) = \frac{1}{(2^n n! \sqrt{\pi})^{1/2}} e^{-\frac{x^2}{2}} H_n(x), \tag{6}$$

such that the mode $n \in \mathbb{N}$ corresponds to the number of zeros of $\varphi_n(x)$. The formalism of number states $|n\rangle$ will be used here for convenience, but note that it is not restrictive to quantum physics, as everything can equivalently be written in terms of Hermite functions $\varphi_n(x)$. As recalled in figure 1, those functions are centered around $x = 0$ (i.e. $\langle x \rangle_n = 0$) that is where the mass term $m$ changes sign, but their spreading increases with $n$ as $\langle x^2 \rangle_n = \frac{1}{2} + n$. Therefore, the lower the mode, the better the confinement.

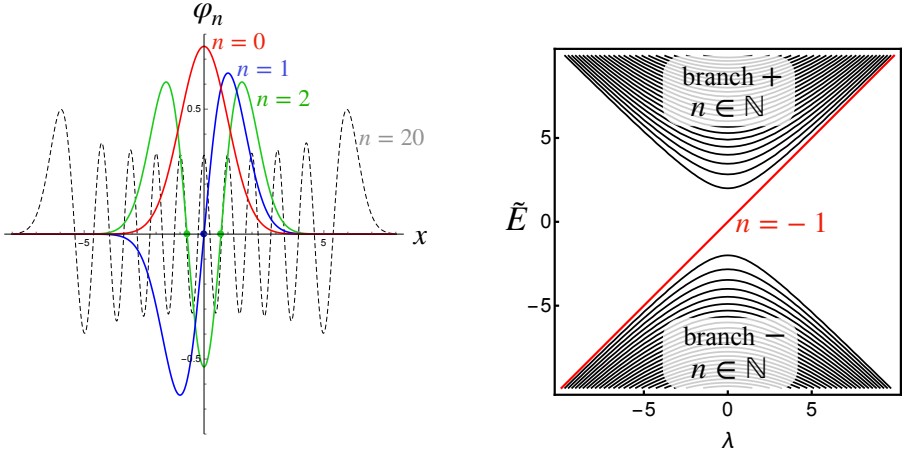

Figure 1: (Left) Spatial dependence of a few Hermite functions. (Right) Spectrum of $\tilde{H}_D$ and of $\tilde{H}_W$. The spectral flow (red) between the two branches correspond to the eigenstate (8).

The Dirac Hamiltonian finally reads

$$\tilde{H}_D = \mathcal{E}\begin{pmatrix} \lambda & \hat{a}^\dagger \\ \hat{a} & -\lambda \end{pmatrix}, \tag{7}$$

where we have introduced the characteristic energy $\mathcal{E} \equiv \sqrt{2}\hbar c/\ell$ and the dimensionless parameter $\lambda \equiv \ell k_y/\sqrt{2}$. The expression of $\tilde{H}_D$, with the operators $\hat{a}$ and $\hat{a}^\dagger$ located on the off diagonal, suggests to look for solutions $|\Psi_n\rangle$ that decompose over number states $|n\rangle$ as

$$\begin{pmatrix} \lambda & \hat{a}^\dagger \\ \hat{a} & -\lambda \end{pmatrix}\begin{pmatrix} \alpha_{n+1}|n+1\rangle \\ \alpha_n|n\rangle \end{pmatrix} = \tilde{E}_n\begin{pmatrix} \alpha_{n+1}|n+1\rangle \\ \alpha_n|n\rangle \end{pmatrix}, \tag{8}$$

where the $\alpha_i$ are normalization coefficients and $\tilde{E}$ is the dimensionless eigenenergy normalized by $\mathcal{E}$. This anzatz yields the discrete spectrum

$$\tilde{E}_n^\pm = \pm\sqrt{\lambda^2 + n + 1} \qquad n \in \mathbb{N}, \tag{9}$$

shown in black in figure 1. It consists of two *branches* $\pm$, separated by a gap around $\tilde{E} = 0$. Each branch is constituted of an infinite number of discrete energy levels $n \in \mathbb{N}$, that correspond to localized modes around $x = 0$. The lower energy mode, $n = 0$, decomposes into two branches $\tilde{E}_0^\pm = \pm\sqrt{\lambda^2 + 1}$ that are associated to the eigenstate

$$\Psi_0(x) = \langle x|\Psi_0\rangle \propto \begin{pmatrix} \langle x|1\rangle \\ \langle x|0\rangle \end{pmatrix} = \begin{pmatrix} \sqrt{2}x \\ 1 \end{pmatrix}e^{-\frac{x^2}{2}}. \tag{10}$$

Actually, the anzatz above does not account for all the solutions of (7). Indeed, an additional solution is suggested by the remarkable structure of $|\Psi_n\rangle$, whose components consist in shifted number states $|n\rangle$ and $|n+1\rangle$ with $n \in \mathbb{N}$. Then, by extrapolating this structure, one can easily check that there exists another lower energy solution that one may somehow abusively call "$n = -1$".[1] This solution reads

$$\Psi_{-1}(x) \propto \begin{pmatrix} \langle x|0\rangle \\ 0 \end{pmatrix} = \begin{pmatrix} 1 \\ 0 \end{pmatrix}e^{-\frac{x^2}{2}} \tag{11}$$

---

[1]This mode is actually called the "zero-mode" in the literature.

and satisfies

$$\begin{pmatrix} \lambda & \hat{a}^\dagger \\ \hat{a} & -\lambda \end{pmatrix} \begin{pmatrix} |0\rangle \\ 0 \end{pmatrix} = \tilde{E}_{-1} \begin{pmatrix} |0\rangle \\ 0 \end{pmatrix}. \tag{12}$$

Unlike the previous solutions for $n \in \mathbb{N}$, this solution yields a *single* branch

$$\tilde{E}_{-1} = \lambda \tag{13}$$

that remarkably transits from the negative energy branch to the positive one when increasing $\lambda$. This is the simplest example of a *spectral flow*, and in the following, $\lambda$ will be referred to as the spectral flow parameter. Put differently, when $\lambda$ is swept from negative to positive values, the branch $-$ looses one state, while the branch $+$ gains one state. Writing $\Delta\mathcal{N}_\pm$ the net number of states which are gained by the branch $\pm$ when varying $\lambda$, one has $\Delta\mathcal{N}_\pm = \pm 1$.

Let us finally notice that the mode that "flows" has several remarkable properties: It is (1) non-dispersive, (2) localized around $x = 0$, that is where $m(x)$ changes sign, (3) fully polarized onto one component only, (4) the only accessible mode in the gap between the two branches $\pm$, (5) it propagates with a positive group velocity for any $k_y$. We will however see later that several of those properties are specific to this simple model. But for now, let us introduce the second example where the same spectral flow appears.

## 2.2 Spectral flow of Landau levels in 3D Weyl semi-metals

Another well-known condensed matter example where a spectral flow arises is that of 3D Weyl semi-metals subjected to a magnetic field [16, 30, 31]. Such materials are characterized by pairs of 2-fold band crossing points, around which the Bloch Hamiltonian can be expanded as massless Weyl Hamiltonians of the typical form [30, 32, 33]

$$H_W = c \begin{pmatrix} \hat{p}_z & \hat{p}_x - \mathrm{i}\hat{p}_y \\ \hat{p}_x + \mathrm{i}\hat{p}_y & -\hat{p}_z \end{pmatrix}. \tag{14}$$

A magnetic field $\mathbf{B} = -B\mathbf{e}_z$, is then applied in the $-z$ direction, and is accounted in this Hamiltonian through the minimal coupling $\hat{\mathbf{p}} \to \hat{\boldsymbol{\pi}} = \hat{\mathbf{p}} + e\mathbf{A}(\mathbf{r})$ with $e$ the elementary electric charge and where the electromagnetic vector potential reads $\mathbf{A}(\mathbf{r}) = B/2(y, -x, 0)$ in the circular gauge. The generalized momentum $\hat{\boldsymbol{\pi}}$ is gauge-invariant, unlike $\hat{\mathbf{p}}$.

Since the system remains invariant by translation in the $z$ direction, one looks for plane wave solutions of wave number $k_z$ along that direction, which amounts to substitute $\hat{p}_z \to \hbar k_z$ in the Hamiltonian. The situation is different in the $(x, y)$ plane: There, the commutator of the generalized momenta $\hat{\pi}_x$ and $\hat{\pi}_y$ does not vanish, but verifies $[\hat{\pi}_x, \hat{\pi}_y] = \mathrm{i}\hbar eB = \mathrm{i}(\hbar/\ell_B)^2$ where the characteristic length $\ell_B \equiv \sqrt{\hbar/eB}$ is called the magnetic length. This means that, unlike $\hat{p}_x$ and $\hat{p}_y$ that do commute, $\hat{\pi}_x$ and $\hat{\pi}_y$ are canonical conjugate variables. This allows us to introduce the annihilation and creation operators

$$\hat{a} = \left(\frac{\ell_B}{\hbar}\right)\frac{\hat{\pi}_x + \mathrm{i}\hat{\pi}_y}{\sqrt{2}} \qquad \hat{a}^\dagger = \left(\frac{\ell_B}{\hbar}\right)\frac{\hat{\pi}_x - \mathrm{i}\hat{\pi}_y}{\sqrt{2}} \tag{15}$$

that we abusively also designate by $\hat{a}$ and $\hat{a}^\dagger$ because they similarly act on the number states as $\hat{a}|n\rangle = \sqrt{n}|n\rangle$, $\hat{a}^\dagger|n\rangle = \sqrt{n+1}|n+1\rangle$ and $[\hat{a}, \hat{a}^\dagger] = 1$.[2] The Weyl Hamiltonian under an axial constant magnetic field reads

$$H_W = \mathcal{E} \begin{pmatrix} \lambda & \hat{a}^\dagger \\ \hat{a} & -\lambda \end{pmatrix}, \tag{16}$$

---

[2]Those ladder operators are actually related to the previous ones given in (4) of the quantum harmonic oscillators in both directions as $\hat{a} = \hat{a}_y - \mathrm{i}\hat{a}_x$, where $\hat{a}_x$ is given by (4), $\hat{a}_y$ is $\hat{a}_x$ after the substitution $x \to y$, and with $\ell = \ell_B$. This is thus essentially a 2D quantum harmonic oscillator in the $x\,y$ plane [34].

where $\mathcal{E} \equiv \sqrt{2}c\hbar/\ell_B$ and $\lambda = k_z\ell_B/\sqrt{2}$. This Hamiltonian is formally exactly the same as that of a 2D Dirac fermion with a mass that varies linearly along the $x$ direction (7), although the characteristic lengths of those two problems have a different origin. The energy spectrum of (16) is therefore already shown in figure 1, and thus displays the same spectral flow as the 2D massive Dirac fermions. Such a spectral flow of Landau levels is currently associated with the condensed matter signature of the chiral anomaly in 3D Weyl semimetals: Assuming the Fermi energy lies around $E = 0$, applying an electric field along the $z$ direction generates a bulk current of electrons. Because Weyl points appear by pairs with opposite Chern numbers, they display opposite spectral flows. The chiral anomaly in that context refers to the fact that, although the number of particles is globally conserved when taking into accounts all the Weyl points, it is not conserved around each Weyl points (also called valleys) separately [17, 18, 35].

It is worth noticing that the ladder operators $\hat{a}$ and $\hat{a}^\dagger$ popped up in the 2D massive Dirac problem and in the 3D magnetic Weyl problem for very different reasons. In the 3D Weyl problem, the key point is the coupling of the electric charge to the magnetic field through the vector potential. In contrast, in the 2D Dirac case, what matters is the existence of an anisotropic mass term that changes sign. This second mechanism can be found in a large variety of classical waves systems whose dynamics is governs by a set of equations of the form $i\partial_t\psi = \hat{H}\psi$ where $\hat{H}$ is an Hermitian and linear operator, and $\psi$ is the vector of the Fourier components of the classical fields involved, such as pressure and velocity fields in fluids or electric and magnetic fields in optics. The very spectral flow shown in figure 1 was actually found in a classical analog of a quantum Chern insulator, namely in 2D gyromagnetic photonic crystals, where the role of the mass term was played by a Faraday coupling [36].

A last remark can be made about the discretization of the spectrum. This discretization can be expected in the 3D Weyl case with a magnetic field, as it corresponds to the emergence of Landau levels. For the 2D massive Dirac fermions, the discretization originates from the assumption of the linear spatial dependence of the mass, and would disappear if the shape of $m(x)$ becomes sharp (step function). But even in that case, the spectral flow between the two branches persists. In other words, it is robust to smooth changes of $m(x)$ provided it changes sign. It is thus natural to account for its robustness with topological tools.

## 2.3 Topology of the symbol for spin $S = 1/2$ fermions

### 2.3.1 Monopole - spectral flow correspondence

The spectral flow $\Delta\mathcal{N}_\pm$ of *operators* such as $H_W$ and $\tilde{H}_D$, has indeed a topological counterpart in the associated *symbol* of such operators, through topological indices called *first Chern numbers* $\mathcal{C}_\pm$. The notions of symbol and of Chern numbers will be introduced in the next sections. In the meantime, let us introduce the simple key relation

$$\Delta\mathcal{N}_\pm = -\mathcal{C}_\pm \tag{17}$$

that constitutes the monopole-spectral flow correspondence. Such a relation is not obvious, but is certainly the simplest illustration of a very abstract theorem, called the Atiyah-Singer index theorem which, under some assumptions, essentially relates an *analytical index* of an operator to a *topological index* of its symbol [26, 37, 38]. This analytical index can then be related to a spectral flow of the operator, implying an equality of the type of equation (17) [39]. In the next sections, we shall give an explicit and computable expression of these indices for a quite general physical model beyond the spin-1/2 case discussed so far.

For now, let us keep focusing on the canonical $S = 1/2$ example as described by the Hamiltonian (7) (or (16)), and remark that in the limit $|\lambda| \gg 1$, the diagonal term, that is proportional to the Pauli matrix $\sigma_z$, becomes dominant. The eigenmodes of the operator $H_W$ or $\tilde{H}_D$

then resemble that of $\sigma_z$, and the typical energy between two consecutive number states $|n\rangle$ and $|n+1\rangle$ in the same branch, $+$ or $-$, tends to zero. There is then a clear separation of energy (or time) scales between the "spin" contribution, that one can associate to a fast motion, and the "oscillator" terms that imply the operators $\hat{a}$ and $\hat{a}^\dagger$, that one can thus associate to a slow motion. With this in mind, it is natural to approximate the system by an "adiabatic" model, where the operators $\hat{a}$ and $\hat{a}^\dagger$ are treated classically. The Hamiltonian deduced from this procedure is precisely the symbol.

As we shall see, the symbol is, in our case, a $2\times 2$ matrix of parameters (the slow variables); the matrix structure being inherited from the two spin components $|\pm\rangle$ (fast variables) that are left untouched. The eigenmodes $|\psi_\pm\rangle$ of the symbol are thus vectors with two complex components, parametrized by the slow variables. These parametrized eigenstates are determined up to a phase, which gives rise to a structure called *fiber-bundle* (see section 2.3.4). So, in the framework of the symbol, the space of fast motion is a fiber bundle over the phase space of slow variables (see e.g. [40]). Fiber bundles are mathematical structures that generically possess a topology which accounts for how the parametrized eigenstates arrange and possibly "twist". In our case, those twists are encoded into the first Chern numbers $\mathcal{C}_\pm \in \mathbb{Z}$.

### 2.3.2 Symbol Hamiltonian $H_S$

The symbol-operator correspondence is routinely used by physicists when replacing by hand a wavenumber $k_x$ by the differential operator $-i\partial_x$. Such a substitution is performed when the invariance by translation is lost in a the corresponding direction, for instance by adding a boundary condition. Conversely, the replacement $-i\partial_x \to k_x$ is performed when invariance by translation is assumed in a linear system governs by differential equations, so that a Fourier transform can be applied. Such substitutions are routinely used by quantum physicists and were formalized by mathematicians in order to make rigorous the mapping between classical and quantum formalisms, i.e. between classical observables defined over phase space (e.g. the momentum $p$) and quantum observables defined by operators acting on a Hilbert space (e.g. $p \to \hat{p} = -i\hbar\partial_x$). In this language, the classical observable $p$ is called the *symbol* and the quantum observable $\hat{p}$ is called the *operator* [41, 42].

We will use this terminology and denote by $\hat{\mathcal{H}}_{\mathrm{op}}$ the *operator Hamiltonian* and by $H_S$ the *symbol Hamiltonian*. Concretely, the previously discussed $H_W$ and $\tilde{H}_D$ in (16) and (7) constitute our operator Hamiltonian $\hat{\mathcal{H}}_{\mathrm{op}}$. Of course, the symbol Hamiltonian is itself also an (Hermitian) operator, since, being a matrix, it acts on vectors in $\mathbb{C}^2$. But its components are entities that commute. In contrast, $\hat{\mathcal{H}}_{\mathrm{op}}$ acts on a Hilbert space that is a product $\mathcal{F} \otimes \mathcal{S}_{1/2}$ of an infinite dimensional Fock space $\mathcal{F}$ spanned by number states $|n\rangle$ with $n \in \mathbb{N}$, with the two-dimensional spin $1/2$ Hilbert space $\mathcal{S}_{1/2}$. In other words, $H_S$ is a matrix of numbers, while $\hat{\mathcal{H}}_{\mathrm{op}}$ is a matrix of operators, which makes its study much more involved. Index theorems provide a way to capture some properties of $\hat{\mathcal{H}}_{\mathrm{op}}$ – namely its spectral flow – thanks to the topological property of a much simpler object, its symbol $H_S$.

Actually the mapping between the symbol and the operator is not unique. In particular, there are many ways to quantize a symbol. However, the symbol carries a topological information that does not depend on the choice of the quantization scheme (this part is called the *principal* or lead symbol). We give a brief introduction to this formalism in the appendix A (more advanced aspects can be found in various textbooks e.g. [41, 42]), and write the quite natural correspondence

$$\lambda_1 + i\lambda_2 \quad \leftrightarrow \quad \hat{a}, \tag{18}$$

$$\lambda_1 - i\lambda_2 \quad \leftrightarrow \quad \hat{a}^\dagger, \tag{19}$$

where $\lambda_1$ and $\lambda_2$ are symbolic notations that designate two canonical conjugate observables in phase space. In the 2D massive Dirac fermion problem (7), they stand for the classical

position $\lambda_1 = x$ and momentum $\lambda_2 = p_x$ observables, while for the 3D Weyl fermion problem (16), they stand for the generalized momenta perpendicular to the applied magnetic field $\lambda_1 = p_x + eA(y)$ and $\lambda_2 = p_y + eA(x)$.

One can then come back to the operator Hamiltonian of our Dirac/Weyl problems and write the correspondence between the symbol Hamiltonian $H_S$ and the operator Hamiltonian $\hat{\mathcal{H}}_{\text{op}}$

$$H_S = \begin{pmatrix} \lambda & \lambda_1 - \mathrm{i}\lambda_2 \\ \lambda_1 + \mathrm{i}\lambda_2 & -\lambda \end{pmatrix} \quad \leftrightarrow \quad \hat{\mathcal{H}}_{\text{op}} = \begin{pmatrix} \lambda & \hat{a}^\dagger \\ \hat{a} & -\lambda \end{pmatrix}. \tag{20}$$

Now that the symbol $H_S$ of $\hat{\mathcal{H}}_{\text{op}}$ has been introduced, one can discuss its topological properties.

### 2.3.3 Degeneracy point of the symbol as a topological defect in 3D parameter space

The purpose of this section is to make concrete what is meant by the *topology* of the symbol $H_S$. Let us denote by $\Lambda = \mathbb{R}^3 \backslash \{\boldsymbol{\lambda}^\star\}$ the *parameter space*, where $\boldsymbol{\lambda}^\star$ designates the degeneracy point(s) of $H_S$, and $\boldsymbol{\lambda} = (\lambda_x, \lambda_y, \lambda_z) = (\lambda_1, \lambda_2, \lambda)$ a point in $\Lambda$. The symbol Hamiltonian (20) we are interested in takes the form

$$H_S = \boldsymbol{\lambda}.\boldsymbol{\sigma}, \tag{21}$$

with $\boldsymbol{\sigma} = (\sigma_x, \sigma_y, \sigma_z)$ the vector of Pauli matrices. Such a $2 \times 2$ Hermitian matrix constitutes the ubiquitous minimal model in condensed matter physics that owns a topological property. As we shall detail in the following, this topological property can be apprehended in two different ways.

Let us start by pointing out that the eigenvalues $E_\pm = \pm|\boldsymbol{\lambda}|$ of $H$ are degenerated at a point $\boldsymbol{\lambda}^\star = 0$. This is actually a generic property of Hermitian matrices, as found by Wigner and von-Neumann (see for instance the discussion of the section VII in [44]) : Indeed, the dimension of the space of Hermitian matrices with a two-fold degeneracy is $n^2 - 3$ (with $n$ the dimension of the matrix), while it is $n^2$ for generic Hermitian matrices. In other words, the codimension of two-fold degenerated Hermitian matrices is 3. This implies that, generically, i.e. in the absence of other constraints imposed by symmetries, 3 parameters must be varied to find a degeneracy between two eigen-energies of a Hermitian matrix. This apparently trivial property actually turns out to play a key role in what follows. In fact, a degeneracy point can be see as a "defect" in parameter space. Similarly to actual topological defects of vector fields in real space (e.g. vortices, dislocations...) one can characterize the degeneracy point by the homotopy property of the map $\boldsymbol{\lambda} : \mathcal{S}_{\boldsymbol{\lambda}} \to \mathcal{S} \subset \mathbb{R}^3$ where $\mathcal{S}_{\boldsymbol{\lambda}}$ is any surface that encloses the degeneracy point. Such maps, between two closed oriented manifolds, (here two surfaces), are classified by an integer-valued number, called the *degree*. The degree of a map is an homotopy invariant, that is, it cannot change value under a continuous deformation of the map. In particular, the two surfaces $\mathcal{S}_{\boldsymbol{\lambda}}$ and $\mathcal{S}$ can conveniently be continuously deformed into spheres $S^2$ in $\mathbb{R}^3$. More generally, the homotopy properties of maps from a sphere $S^n$ to a sphere $S^m$ are classified by homotopy groups noted $\pi_n(S^m)$. The search for homotopy groups of spheres is in general a difficult open problem in mathematics. However, there is a particular situation where the result is known, which is $\pi_n(S^n) = \mathbb{Z}$, and thus is particular $\pi_2(S^2) = \mathbb{Z}$. This means that point-like topological defects of 3D vector fields are classified by integers. These integers are precisely the degree, which, geometrically, counts how many times the image of $\mathcal{S}_{\boldsymbol{\lambda}}$ by the map $\boldsymbol{\lambda} \in \mathcal{S}_{\boldsymbol{\lambda}} \to h(\boldsymbol{\lambda}) \in \mathcal{S}$ covers $\mathcal{S}$, or equivalently how many times the normalized map $\mathcal{S}_{\boldsymbol{\lambda}} \to \mathbf{n} \equiv h/\mathbf{h} \in S^2$ wraps the unit sphere. In our simple example, where $h$ is identity, the degree is 1. In the following, we shall discuss much richer situations where the degree can take arbitrary values. The vector field around topological defects characterized

with different degrees cannot be smoothly deformed into each other. This characterizes the topological robustness of the defect.

### 2.3.4 Phase singularity of the eigenstates

Characterizing the degeneracy point of the symbol Hamiltonian with the degree of a map amounts to ignoring the spinorial structure of the problem, carried by the Pauli matrices in (21). Indeed, $H_S$ is a matrix and a subtle topological property is hidden in its normalized eigenvectors $\psi_\pm$

$$|\psi_+\rangle = \frac{e^{i\chi_+}}{\sqrt{2|\boldsymbol{\lambda}|(|\boldsymbol{\lambda}| - \lambda_z)}} \begin{pmatrix} \lambda_x - i\lambda_y \\ |\boldsymbol{\lambda}| - \lambda_z \end{pmatrix}, \tag{22}$$

$$|\psi_-\rangle = \frac{e^{i\chi_-}}{\sqrt{2|\boldsymbol{\lambda}|(|\boldsymbol{\lambda}| + \lambda_z)}} \begin{pmatrix} \lambda_x + i\lambda_y \\ -|\boldsymbol{\lambda}| - \lambda_z \end{pmatrix}, \tag{23}$$

where $\chi_\pm$ is an arbitrary phase that depends on $\boldsymbol{\lambda}$. For any value of $\boldsymbol{\lambda}$, there is a gauge freedom for the choice of $\chi_\pm$, as usual in quantum mechanics. This means that for a given $\boldsymbol{\lambda}$, there is not a single state that describes the system, but a continuous family of states that differ from each other by a phase factor. This is, in a sense, a multi-valuation of the solution. In contrast, the projectors $\Pi_\pm \equiv |\psi_\pm\rangle\langle\psi_\pm|$ do not depend on the gauge and are thus always single-valued. Once the gauge is chosen, the eigenstate should be single-valued as well. This is however not always the case, as one can suspect from (23) when $\boldsymbol{\lambda}$ is aligned along the $z$ direction, i.e. for $\pm\lambda_z = \pm|\boldsymbol{\lambda}|$.

To see it more clearly, let us use the spherical coordinates with $\lambda_x = |\boldsymbol{\lambda}|\sin\theta\cos\phi$, $\lambda_y = |\boldsymbol{\lambda}|\sin\theta\sin\phi$ and $\lambda_z = |\boldsymbol{\lambda}|\cos\theta$, and parametrize the spinor on the Bloch sphere, where it becomes explicit that the eigenstates only depend on $\mathbf{n}$ (up to the gauge choice) :

$$|\psi_+\rangle = e^{i\chi_+} \begin{pmatrix} \cos\frac{\theta}{2}e^{-i\phi} \\ \sin\frac{\theta}{2} \end{pmatrix} \quad \text{and} \quad |\psi_-\rangle = e^{i\chi_-} \begin{pmatrix} \sin\frac{\theta}{2}e^{-i\phi} \\ -\cos\frac{\theta}{2} \end{pmatrix}. \tag{24}$$

There, setting $\mathbf{n} = n_z > 0$ or equivalently $\theta = 0$ (north pole), the eigenstate $\Psi_+$ becomes

$$|\psi_+\rangle \overset{\theta=0}{=} e^{i\chi_+} \begin{pmatrix} e^{-i\phi} \\ 0 \end{pmatrix} \quad \text{(at the north pole of the Bloch sphere)}, \tag{25}$$

which is multi-valued even after the gauge has been fixed, unless we make the particular gauge choice $e^{i\chi_+} = e^{i\phi}$. This fixes the multi-valued of the eigenstate, but only at the north pole. Indeed, in this gauge, the eigenstate becomes multivalued at the south pole $\theta = \pi$

$$|\psi_+\rangle \overset{\theta=\pi}{=} \begin{pmatrix} 0 \\ e^{i\phi} \end{pmatrix} \quad \text{(with } e^{i\chi_+} = e^{i\phi}) \quad \text{(at the south pole of the Bloch sphere)}. \tag{26}$$

Actually, the multi-valuedness of the eigenstates cannot be fixed by a *global* gauge choice. Or say otherwise, there does not exist a smooth function $\chi$ that makes the eigenstate single-valued everywhere on the Bloch sphere. One can only choose *locally* a gauge that makes the eigenstate single-valued, namely

$$\text{south gauge:} \quad \chi = 0 \quad |\psi_+^S\rangle = \begin{pmatrix} \cos\frac{\theta}{2}e^{-i\phi} \\ \sin\frac{\theta}{2} \end{pmatrix} \quad \text{for } \theta \neq 0, \tag{27}$$

$$\text{north gauge:} \quad \chi = \phi \quad |\psi_+^N\rangle = \begin{pmatrix} \cos\frac{\theta}{2} \\ \sin\frac{\theta}{2}e^{i\phi} \end{pmatrix} \quad \text{for } \theta \neq \pi. \tag{28}$$

Thus, the eigenstates are only piece-wise single-valued, and related by a gauge transformation

$$|\psi_+^N\rangle = e^{i\phi} |\psi_+^S\rangle \,. \tag{29}$$

This reflects a topological property of the eigenstates, as their phase cannot be defined smoothly globally: it has to have a singularity somewhere, similarly to a vortex. This topological property is inherently related to the gauge freedom, and is mathematically described by the theory of fiber-bundles [26,43].

Fiber bundles constitute a rigorous construction of a product of spaces. Locally, the product of two spaces, say a segment $[-1,1]$ with a circle $S^1$, is given by the usual cartesian product. But globally, this does not necessarily hold since a twist may occur. In that case, the fiber-bundle is said to be *topologically non-trivial*. This is the celebrated example of the Moebius strip that differs by a twist from the cylinder. These two structures both result from the product of the segment with the circle, but the Mobius strip can only be seen as the cartesian product $[-1,1] \times S^1$ locally. Fiber bundles are in particular useful to make meaningful the notion of parametrized vector space. The vector space (here the segment) is called the fiber, while the space that parametrizes the fiber (here the circle) is called the base space.

Another intuitive example of a fiber-bundle is the family of tangent vector spaces to the sphere $S^2$. In that case, the vector space is the tangent plane to the sphere and the base space is the sphere itself. The tangent vector bundle is thus the collection of all tangent vectors at any point of the sphere. This vector bundle is also topological, and its twist can be seen in the tangent vector field to the sphere, that necessarily has two vortices, which are precisely points where the phase (i.e. the angle of the tangent vectors with respect to an arbitrary direction) is multi-valued. The position of these singularities depends on the gauge choice, in the language of physicists, or of the section of the fiber bundle as mathematicians would say. But the total vorticity, equals to 2 in that example, cannot change. This is a topological invariant of this fiber bundle, and this result is known as the hairy ball theorem.

The fiber bundles we are interested in resemble this hairy ball : the base space is also the sphere $S^2$ but the fibers (the hairs) are complex vector spaces made of all the eigenstates $\psi$ that only differ by a phase (or gauge choice), that is

$$F_\pm(\boldsymbol{\lambda}) = \{e^{i\chi_\pm(\boldsymbol{\lambda})}\psi_\pm(\boldsymbol{\lambda}), e^{i\chi_\pm(\boldsymbol{\lambda})} \in U(1)\} \,. \tag{30}$$

It is said that the fibers $F_\pm(\boldsymbol{\lambda})$ define the *equivalent class* of the states $\psi_\pm(\boldsymbol{\lambda})$ as they have the same projector and thus describe the same physical state. For each mode $\pm$, the fiber bundle (of eigenvector bundle) of interest consists in the continuous collection of such $U(1)$-fibers $F_+$ or $F_-$, where the base space is a sphere that encloses the degeneracy point. A two-fold degeneracy point is thus associated with two eigenvector bundles.

There are many other fiber bundles, of different dimensions, of real or complex nature, with group structures different from $U(1)$, but this simple example already englobes many interesting physical situations.

### 2.3.5 Chern number from the Berry curvature as an obstruction to Stokes theorem

To express the topological invariants of fiber bundles, one needs now to introduce some definitions of differential geometry. The language of differential forms is particularly convenient to reveal the topological and geometrical structures of physical systems. Of main importance to characterize the eigenstates $|\psi_n(\boldsymbol{\lambda})\rangle$ of a quantum Hamiltonian that is smoothly parametrized in $\Lambda$, or actually any eigenstates of a parametrized linear Hermitian eigenvalues problem of the form

$$H(\boldsymbol{\lambda})|\psi_n(\boldsymbol{\lambda})\rangle = E_n(\boldsymbol{\lambda})|\psi_n(\boldsymbol{\lambda})\rangle \tag{31}$$

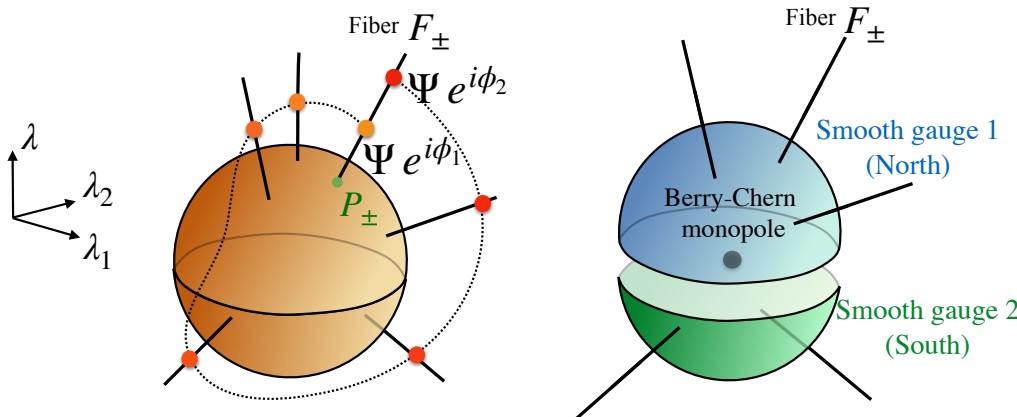

Figure 2: (a) $U(1)$ fiber-bundle as a continuous collection of fibers $F$ over the sphere defined in parameter space $\lambda$. Two states in the same fiber only differ by a phase; they thus have the same projector onto the base space. The impossibility to define smoothly this phase over the base space is a manifestation of the topological property of the fiber bundle, encoded into the Chern number. (b) This Chern number is the quantized flux through the sphere of the Berry curvature generated by the degeneracy point, called Berry monopole. The existence of a non-zero Chern number imposes the 1-form Berry connection to be defined smoothly only piecewise.

that one can encounter in particular in wave physics, are geometrical tools introduced by Berry and Simon, called *Berry connection* and *Berry curvature* [45, 46]. The Berry connection is a 1-form defined from each parametrized eigenstate as[3]

$$
\begin{aligned}
\mathcal{A}^{(n)}(\lambda) &\equiv i \langle \psi_n | d | \psi_n \rangle \\
&= i \langle \psi_n | \frac{\partial}{\partial \lambda_j} | \psi_n \rangle d\lambda_j \\
&\equiv A_j^{(n)}(\lambda) d\lambda_j \,,
\end{aligned}
\tag{32}
$$

where the sum over the $j$ index is implicit.[4] The one-form Berry connection is not gauge invariant : when a $U(1)$-gauge transformation is performed on a state as

$$
|\psi\rangle \rightarrow e^{i\chi(\lambda)} |\psi\rangle = |\tilde{\psi}\rangle \,,
\tag{33}
$$

then the Berry connection transforms as

$$
\mathcal{A}^{(n)} \rightarrow \tilde{\mathcal{A}}^{(n)} = i \langle \tilde{\psi}_n | d | \tilde{\psi}_n \rangle
\tag{34}
$$

$$
= -d\chi + \mathcal{A}^{(n)} \,.
\tag{35}
$$

This is very analogous to a vector potential in electromagnetism, where moreover the language of differential forms automatically allows us to work in arbitrary dimension, where the notion of e.g. the *curl* becomes obscure. Similarly to electromagnetism, one can introduce the equivalent of a magnetic field, called the Berry curvature, as

$$
\mathcal{F}^{(n)}(\lambda) \equiv d\mathcal{A}^{(n)}(\lambda)
\tag{36}
$$

$$
= \frac{\partial A_j^{(n)}}{\partial \lambda_i} d\lambda_i \wedge d\lambda_j \,,
\tag{37}
$$

---

[3]This definition fits the one of Berry and Simon but differs by a minus sign from the one mostly encountered in solid states physics when dealing with Bloch bands.

[4]We keep the upper index to denote the band index.

where the implicit sum now runs over $i$ and $j$. Because of the anti-symmetry $\mathrm{d}\lambda_i \wedge \mathrm{d}\lambda_j = -\mathrm{d}\lambda_j \wedge \mathrm{d}\lambda_i$ of the wedge product $\wedge$, one has

$$\mathcal{F}^{(n)}(\boldsymbol{\lambda}) = \frac{1}{2}\left(\frac{\partial}{\partial\lambda_i}A_j^{(n)} - \frac{\partial}{\partial\lambda_j}A_i^{(n)}\right)\mathrm{d}\lambda_i \wedge \mathrm{d}\lambda_j \tag{38}$$

$$\equiv \frac{1}{2}F_{ij}^{(n)}\mathrm{d}\lambda_i \wedge \lambda_j, \tag{39}$$

where the second equality defines the coefficients $F_{ij}^{(n)}(\boldsymbol{\lambda})$ of the 2-form Berry curvature $\mathcal{F}^{(n)}$. Those are left invariant under the $U(1)$-gauge transformation (33). Consistently, they are observable quantities that manifest themselves for instance in the semi-classical equations of motion of wavepackets travelling in spatially slowly varying media, and in the adiabatic evolution of quantum states [47–52].

Mathematically, this gauge invariance of the Berry curvature is straightforwardly inferred from the formalism of differential forms. Indeed, at this stage, we have introduced the Berry curvature as a 2-form that derives from of the 1-form Berry connection by taking the exterior derivative d. Such a construction is called an *exact* form (see appendix B), and it is known as a theorem that the exterior derivative of an exact form vanishes. Therefore, under the gauge transformation (33), the Berry curvature transforms as $\mathcal{F} \to \mathcal{F} - \mathrm{d}^2\chi = \mathcal{F}$.

A very subtle and crucial point is that the Berry curvature may actually not be an exact form for every $\boldsymbol{\lambda} \in \Lambda$. The topology arises precisely in such a situation, as the obstruction to define *globally* the Berry curvature as $\mathcal{F}(\boldsymbol{\lambda}) = \mathrm{d}\mathcal{A}(\boldsymbol{\lambda})$ for every $\boldsymbol{\lambda}$. In that case, this equality is only valid *locally*, i.e. on sub-domains of $\Lambda$, and $\mathcal{A}^{(n)}(\boldsymbol{\lambda})$, (and consequently $\mathrm{d}\mathcal{A}^{(n)}(\boldsymbol{\lambda})$), is only piecewise well-defined. This is precisely the case when the phase of the eigenstates, such as $\psi_\pm$ (see (24)) of the symbol Hamiltonian $H_S$ for the spin $S = 1/2$, cannot be smoothly defined over $\Lambda$.

As we shall see, the obstruction to define globally the phase of the eigenstates, or equivalently their Berry curvature, is captured quantitatively by an integer-valued topological number, called the *first Chern number*, expressed as

$$\mathcal{C}_n \equiv \frac{1}{2\pi}\int_{S^2}\mathcal{F}^{(n)} \in \mathbb{Z}. \tag{40}$$

Again, there is a strong analogy with electromagnetism, where the expression (40) of the Chern number is given by the flux of the Berry curvature. This analogy can be pushed further by expressing the coefficients $F_{ij}^{(n)}$ of the Berry curvature as

$$F_{ij}^{(n)}(\boldsymbol{\lambda}) = \mathrm{i}\sum_{m\neq n}\frac{\langle\psi_n|\partial_{\lambda_i}H|\psi_m\rangle\langle\psi_m|\partial_{\lambda_j}H|\psi_m\rangle - \langle\psi_n|\partial_{\lambda_j}H|\psi_m\rangle\langle\psi_m|\partial_{\lambda_i}H|\psi_m\rangle}{(E_n(\boldsymbol{\lambda})-E_m(\boldsymbol{\lambda}))^2}, \tag{41}$$

which is obtained by differentiating (31) and inserting it into (38). This formula has a practical interest, and we shall use it in a next section to compute analytically the Berry curvature for a general spin-$S$ model. For now, let us stress that the denominator of (41) reveals that the amplitude of the Berry curvature of a band $n$ increases as this band gets closer to the other bands $m$ in parameter space $\Lambda$. In particular, the Berry curvature diverges at a degeneracy point (provided the numerator is regular). For that reason, degeneracy points (or band crossing points) play the role of point-like source of the Berry curvature. The Chern number number (40) can then be understood as the (topological) charge of this monopole, with the (close) surface of integration $S^2$ enclosing the degeneracy point $\lambda^\star$ in $\Lambda$ space. Analogies with the magnetic monopole are naturally often made, since the quantization of the electric charge

was found by Dirac to be a consequence of the obstruction to smoothly define the vector potential that couples to the quantum wavefunction around such monopoles. Note however that a $N$-fold band crossing points – that we dub here Berry-Chern monopoles – imply $N$ bands, and thus $N$ fiber bundles, each of them being characterized with a Chern number $\mathcal{C}_n$. Such degeneracy points are thus associated with $N$ topological charges rather than a single one. In the simple case of two-band crossing points, which are the most often encountered in the literature, such as Weyl points, this terminology of topological charge is currently used without too much ambiguity, because the two Chern numbers are necessarily opposite. This follows from the important property of the Berry curvature that satisfies

$$\sum_n F_{ij}^{(n)}(\boldsymbol{\lambda}) = 0\,, \tag{42}$$

where the sum runs over all the bands $n$. This property, that can be derived from the expression (41), yields the important relation on the Chern numbers

$$\sum_n \mathcal{C}_n = 0\,. \tag{43}$$

Thus, an $N$-fold degeneracy point $\boldsymbol{\lambda}^\star \in \Lambda$ is characterized by $N$ Chern numbers $\mathcal{C}_n$ satisfying this constraint. We refer here to such points as Berry-Chern monopoles.

To see how the Chern number (40) captures the obstruction to smoothly define the eigenstates (or to have the Berry curvature as an exact 2-form) globally, let us apply Stokes theorem (B.5)

$$\int_{S^2} \mathcal{F} = \int_{S^2} \mathrm{d}\mathcal{A} = \int_{\partial S^2} \mathcal{A}\,, \tag{44}$$

(where we drop the band index when unecessary). Since the sphere $S^2$ is a closed surface, it does not have a boundary (i.e. $\partial S^2 = 0$), so that the right hand side member of this equality always vanishes. This result is only valid if $\mathcal{F} = \mathrm{d}\mathcal{A}$ everywhere on the domain of integration. If, on the other hand, the equality $\mathcal{F} = \mathrm{d}\mathcal{A}$ only holds piecewise, then Stokes theorem cannot be used over $S^2$. For that reason, it is said that the Chern number is an obstruction to Stokes theorem. Let us illustrate this point with the eigenstate $\psi_+$ of the spin $1/2$ problem. We saw in Eqs. (27) and (28) that such a state can only be smoothly defined piecewise, in different gauges, and therefore so is the Berry connections

$$\mathcal{A} = \mathrm{i}\langle\psi|\,\partial_\theta\,|\psi\rangle\,\mathrm{d}\theta + \mathrm{i}\langle\psi|\,\partial_\phi\,|\psi\rangle\,\mathrm{d}\phi \tag{45}$$

leading to

$$\text{south gauge:} \quad \chi = 0 \qquad \mathcal{A}_S^{(+)} = \cos^2\frac{\theta}{2}\mathrm{d}\phi \qquad\qquad \text{for } \theta \neq 0\,, \tag{46}$$

$$\text{north gauge:} \quad \chi = \pi \qquad \mathcal{A}_N^{(+)} = -\sin^2\frac{\theta}{2}\mathrm{d}\phi \qquad\qquad \text{for } \theta \neq \pi\,. \tag{47}$$

The introduction of the Berry curvature as the derivative of the Berry connections $\mathcal{F} = \mathrm{d}\mathcal{A}$ must accordingly be performed over the corresponding domains, associated with the two different gauge choices. The Chern number can then be computed via Stokes theorem by cutting the sphere on different patches where this equality makes sense, for instance over north hemi-

sphere (NH) and south hemisphere (SH) respectively, as illustrated in figure 2 :

$$\mathcal{C}_+ = \frac{1}{2\pi} \int_{S^2} \mathcal{F}_+ \tag{48}$$

$$= \frac{1}{2\pi} \int_{\mathrm{NH}} \mathcal{F}_+ + \frac{1}{2\pi} \int_{\mathrm{SH}} \mathcal{F}_+ \tag{49}$$

$$= \frac{1}{2\pi} \int_{\mathrm{NH}} \mathrm{d}\mathcal{A}_+^N + \frac{1}{2\pi} \int_{\mathrm{SH}} \mathrm{d}\mathcal{A}_+^S, \tag{50}$$

so that Stokes theorem can be safely used to get

$$\mathcal{C}_+ = \frac{1}{2\pi} \int_{\partial \mathrm{NH}} \mathcal{A}_+^N + \frac{1}{2\pi} \int_{\partial \mathrm{SH}} \mathcal{A}_+^S, \tag{51}$$

where the boundaries $\partial \mathrm{NH}$ and $\partial \mathrm{SH}$ are any common path $\gamma$ that does not cross a pole, and are oriented in opposite directions for the two domains, so that

$$\mathcal{C}_+ = \frac{1}{2\pi} \int_\gamma \mathcal{A}_+^N - \mathcal{A}_+^S. \tag{52}$$

The two connections involved are defined on domains where the eigenstates $|\psi_+^N\rangle$ and $|\psi_+^S\rangle$ are related by the gauge transformation (29). According to (35), the Berry connections $\mathcal{A}_+^N$ and $\mathcal{A}_+^S$ inherit this relation as

$$\mathcal{A}_+^N = \mathcal{A}_+^S - \mathrm{d}\phi \tag{53}$$

the Chern number is directly inferred as

$$\mathcal{C}_+ = \frac{1}{2\pi} \int_0^{2\pi} -\mathrm{d}\phi = -1. \tag{54}$$

Conversely, one finds $\mathcal{C}_- = +1$ if $\psi_-$ is considered instead of $\psi_+$, consistently with the constraint (43). The Chern numbers $\mathcal{C}_\pm$ of the symbol Hamiltonians $H_S$ thus satisfy the monopole-spectral flow correspondence (17) for the spin 1/2 band crossing problem.

Let us conclude this section by a few remarks on the computation of the Chern numbers. First one can check the value of the Chern number by inserting the expressions of the Berry connections $\mathcal{A}_S^{(+)}$ and $\mathcal{A}_N^{(+)}$ in (52) and then taking the path $\gamma$ as the equator of $S^2$, that is $\theta = \pi/2$. In passing, one can also check that these two different expressions for the Berry connection indeed yield the same Berry curvature $\mathcal{F}^+ = -\frac{1}{2} \sin \theta \mathrm{d}\theta \wedge \mathrm{d}\phi$ whose expression is valid for any $\theta$. A direct integration of this quantity over $S^2$ in spherical coordinates also consistently yields $\mathcal{C}_+ = -1$. However, let us stress that the calculation of the Chern number presented above actually did not require the explicit computation of the Berry connection, neither that of the Berry curvature. It only lies on the gauge transformation between the different patches where the eigenstates are well-defined. This can be slightly rewritten with

$$|\psi\rangle \to |\tilde{\psi}\rangle = \mathrm{e}^{\mathrm{i}\chi(\lambda)} |\psi\rangle, \tag{55}$$

$$\mathcal{A} \to \mathcal{A} + \mathrm{i}\mathrm{e}^{-\mathrm{i}\chi} \mathrm{d}\mathrm{e}^{\mathrm{i}\chi}, \tag{56}$$

where $\mathrm{e}^{\mathrm{i}\chi} \in U(1)$ is called the *transition function*, and the Chern numbers then correspond to the winding numbers of this function

$$\mathcal{C} = \frac{1}{2\pi} \int_\gamma \mathcal{A} - \tilde{\mathcal{A}} = -\frac{1}{2\pi\mathrm{i}} \int_\gamma \mathrm{e}^{-\mathrm{i}\chi} \mathrm{d}\mathrm{e}^{\mathrm{i}\chi}. \tag{57}$$

## 2.4 Spectral flow from the Berry-Chern monopole of the symbol : a simple methodology

### 2.4.1 Predicting a spectral flow from 2D and 3D dispersion relations

At this stage, we have verified the monopole - spectral flow correspondence (17) for spin 1/2 models with a linear band crossing point. It is worth stressing that the correspondence (20) between the symbol and the operator allows one to predict a spectral flow, that is the relevant physical information, from the symbol Hamiltonian which is a much simpler object to manipulate. This provides us with a powerful tool, especially because in the cases we address here, the symbol's spectrum is nothing but the dispersion relation of the homogeneous system. A generic method to predict a spectral flow therefore consists in looking for a band crossing in the dispersion relation that appears as point in a 3D parameter space. This is actually the standard situation for two-band crossings in the absence of symmetry, according to Wigner-von-Neumann theorem. The computation of the Chern number, detailed in the previous section, may seem technical to non familiar readers. But, since generic band crossings only involve two bands, the effective Hamiltonian describing locally such crossings is the spin 1/2 Hamiltonian we have discussed, up to unitary transformations, so that the Chern numbers can only be $+1$ or $-1$. Therefore, it is sufficient to find out a band crossing point with a linear dispersion relation (which is an easy task) in order to predict a spectral flow of one mode (which is much more involved to find in full generality). If the band-crossing point involves more bands and/or a nonlinear dispersion relation, then a careful treatment of the Chern numbers must be done. This is the purpose of the section 3.

To guarantee the emergence of a spectral flow, one needs to turn the symbol problem, that describes an homogeneous system, into an inhomogeneous one described by the operator problem. The examples discussed in the first section provides two physical recipes to perform such a transformation, depending on the dimension of the system. Those two methods remain valid even beyond two-fold band crossing points with a linear dispersion relation.

If, on the one hand, the physical problem is three-dimensional, (i.e. the dispersion relation $E(\mathbf{k})$ is such that $\mathbf{k}$ is three-dimensional), then one must look for a physical mechanism that turns two of the three components of the wave number $\mathbf{k}$ into canonical conjugate variables. This is what happens when one uses the minimal coupling to account for the application of a magnetic field in Weyl semi-metals. The spectral flow then consists in the transfer of the lowest Landau level leading to a bulk propagating mode in the direction parallel to the magnetic field. This propagating mode leads to a current when an electric field is applied (provided the mode crosses the Fermi energy), which is interpreted as a signature of the chiral anomaly.

If, on the other hand, the physical system is two-dimensional, (i.e. the dispersion relation $E(\mathbf{k})$ is such that $\mathbf{k}$ is two-dimensional), then one must look for a third parameter to be tuned such that it changes sign when a gap between wavebands closes. This is the role of the mass term in the 2D massive Dirac problem. Note that the symmetries are a good guide to reveal such a parameter: Indeed, degeneracy points generically occur when a symmetry is exceptionally restored in the problem, such as inversion or time reversal. The parameter playing the role of the mass term must thus break such a symmetry. The operator Hamiltonian is then obtained when considering that the mass term becomes a smooth function of space $m(x)$, and a spectral flow is expected along the perpendicular direction $k_y$ if $m(x_0)$ vanishes at some $x_0$. This necessary vanishing of $m(x)$ in the inhomogeneous problem, is the dual (i.e. operator) counter-part of the vanishing of the parameter $m$ in the homogeneous problem, which is at the source of the Berry-Chern monopole. The physical interpretation of that spectral flow is that of an uni-directional interface mode, i.e. a wave confined in the perpendicular direction $(x)$ to the line $m(x_0) = 0$ but that propagates along the interface $(y)$ in one direction only.

SciPost Phys. Lect.Notes 39 (2022)

### 2.4.2 Bulk-interface correspondence in 2D

The trapped unidirectional modes of the spectral flow in 2D inhomogeneous systems look very much like the topological chiral edge states propagating along the boundary of two-dimensional quantum Hall like systems, i.e. Chern insulators.

Let us recall that Chern insulators are band insulators whose energy (or frequency) Bloch bands $n$ are characterized by a Chern number $C_n$. Those *band* Chern numbers are given by the integral of the Berry curvature of each Bloch bundle (the fiber bundle of Bloch eigenstates) over the 2D Brillouin zone. The periodicity in both directions of the 2D Brillouin zone (BZ) making it equivalent to a 2-torus $T^2$ torus, the band Chern number reads

$$C_n = \frac{1}{2\pi} \int_{BZ \sim T^2} \mathcal{F}^{(n)}(\mathbf{k}).$$
(58)

In contrast, the surface of integration of the Berry curvature we considered for the monopole Chern numbers $\mathcal{C}_n$ encloses a degeneracy point in $\mathbb{R}^3 \backslash \{\boldsymbol{\lambda}^\star\}$ space.

The band Chern number (58) characterizes the obstruction to smoothly define the phase of the Bloch eigenstates over the Brillouin zone. Time-reversal symmetry must be broken for the band Chern number to be non-zero. A non-zero band Chern number of a (homogeneous) periodic system, guarantees the existence of a unidirectional (chiral) edge mode that manifests itself as a spectral flow in the non-periodic system with boundaries. This is the celebrated bulk-edge correspondence [53–55]. There is a close connection between the bulk-edge correspondence and the monopole - spectral flow correspondence for 2D systems, since an interface between two distinct Chern insulators can be described by a single Hamiltonian operator with a varying mass term whose sign controls the topological transitions between the two band insulators. Let us illustrate this point with a canonical model for Chern insulators: the Haldane model [28]. It consists of a honeycomb lattice model, with real nearest neighbours couplings (essentially the tight-binding model of graphene), plus complex hopping $e^{i\phi}$ terms to second nearest neighbours, thus breaking time-reversal symmetry, and a on-site staggered potential $\Delta$ that breaks inversion symmetry (see figure 3 (a)). This two-band model displays three topologically distinct phases with $C_\pm = 0, \pm 1, \mp 1$, depending on the competition between the two gap opening mechanisms controlled by $\phi$ and $\Delta$, and are separated by gap closing points (in orange and blue in figure 3 (b)).

Those gap closing points occur around one of the two valleys $K$ and $K'$ of the graphene dispersion relation (figure 3 (b)). At low energy (i.e. around $E = 0$), this gap-opening management is controlled by two valley dependent mass terms $m_K$ and $m_{K'}$. The topological transitions between two distinct Chern phases are thus encoded into a low energy description around each valley $K$ or $K'$. In that limit, the Haldane model reduces to two 2D valley dependent Dirac Hamiltonians with a distinct mass $m_K$ or $m_{K'}$, that depend on the two parameters of the lattice model, $\Delta$ and $\phi$. These Dirac Hamiltonians can be interpreted as symbol Hamiltonians parametrized respectively in $(\delta k_x, \delta k_y, m_{K^{(')}})$ parameter spaces with $\delta \mathbf{k}$ a small expansion around the valley $K^{(')}$. Those symbol Hamiltonians have a Berry-Chern monopole that coincides with the transition between the two insulators, i.e. for $m_{K^{(')}} = 0$. The interface problem of a Chern insulator with a trivial one is therefore captured by Hamiltonian operators $\hat{\mathcal{H}}_{\mathrm{op}}$ when taking a mass varying term $m_K$ or $m_{K'}$ (path 1 in figure 3), say in the $x$ direction. The values $\mathcal{C}_\pm = \mp 1$ of those Berry-Chern monopoles agree with the difference of bulk Chern numbers of the Bloch bands between the two insulators, and the associated spectral flow corresponds to the interface chiral mode.

Interestingly, the monopole-spectral flow correspondence can also be used to describe a transition between two trivial insulators (path 2 in figure 3). In that case, the bulk-edge correspondence predicts no interface state, because the band Chern numbers $C_\pm$ of the Bloch

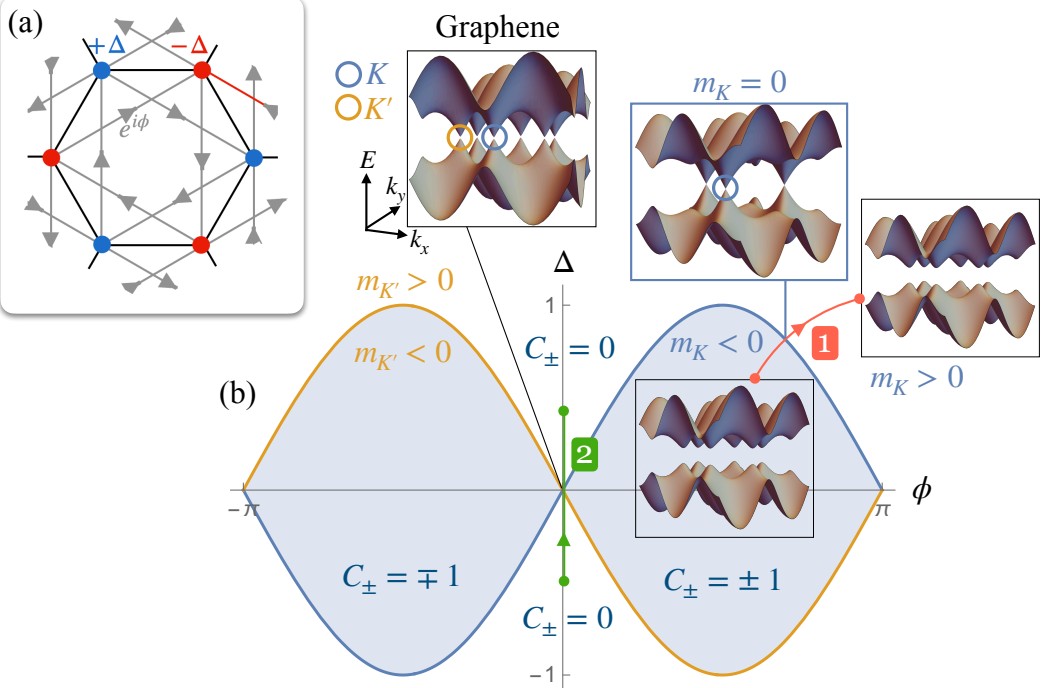

Figure 3: (a) Haldane model on the honeycomb lattice. (b) Phase diagram of the Haldane model. The topologically distinct insulating phases are characterized by the bulk Chern number $C_\pm$ of each Bloch band over the Brillouin zone. They are separated by a gap closing point at either $\mathbf{k} = K$ or $\mathbf{k} = K'$. The gap amplitude is controlled by valley-dependent mass terms $m_K$ and $m_{K'}$. Along the path 1, the gap closes at a single valley.

bands of both insulators are zero. However, the parameter $\Delta$, that breaks inversion symmetry, is varied when going from one insulator to the other one, and the gap closes when this parameter changes sign. At the gap closing point, the mass terms associated to each valley $m_K$ and $m_{K'}$ vanish and one recovers the usual graphene model with two Dirac points. The monopole-spectral flow correspondence is *local* in parameter space, and thus applies for each band crossing point, and not to the bulk Bloch bands at fixed $\Delta$. Chern numbers $\mathcal{C}_\pm^K$ and $\mathcal{C}_\pm^{K'}$ of the Berry-Chern monopoles can thus be computed for each valley, like previously, and are actually opposite to each other. It follows that the interface problem, that is when $\Delta$ is varied in space and changes sign, manifests a double spectral flow which is opposite in each valley, as illustrated by the numerical spectrum of the Haldane model for a sharp interface in figure 4. This is the *(quantum) valley Hall effect*, and the Berry-Chern monopole is reminiscent of the so-called *valley* Chern number [56, 57]. This phenomenon triggered particular interest in classical analogs of topological materials due to the experimental ease to break inversion symmetry [10, 11, 58]. Note that because the monopole-spectral flow correspondence is local in parameter space, it actually does not require the system to be an insulator: gap inversion can be performed in a given valley, while the bands always touch somewhere else in the Brillouin zone, giving rise to topological spectral flows in (semi-)metallic like systems, where the band Chern numbers are ill defined [59].

Finally, we should also stress that the monopole-spectral flow correspondence is particularly suitable to investigate the existence of unidirectional waves in continuous media [15, 60–62]. In the absence of a lattice, and thus without a Brillouin zone, there is no natural close surface to integrate the Berry curvature over, and therefore, in general, no well-defined band

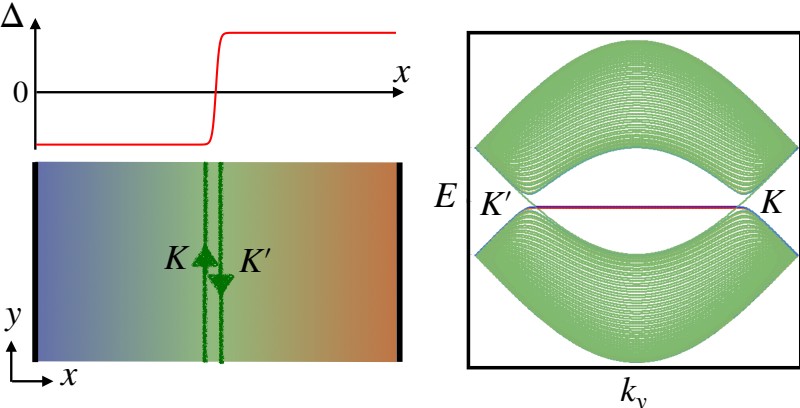

Figure 4: Numerical spectrum of a sharp interface between Haldane models with opposite staggered potential $\Delta$, in a strip geometry, finite in the $x$ direction and infinite in the $y$ direction. the color represent the mean position of the eigenstate in the $x$ direction. The two valleys host opposite spectral flows, corresponding to valley polarized counter propagating modes localized in the middle of the strip where $\Delta$ changes sign. In contrast, the two flat bands edge states do not contribute to the spectral flow.

Chern number. Indeed, the Hamiltonian describing the homogeneous system depends then on the wavenumber $\mathbf{k} \in \mathbb{R}^2$ in the (open) plane. Although the plane can be compactified into a sphere, such Hamiltonians in the continuum do not always have, in general, the required regularization properties at infinity such that the fiber bundle can be properly defined over this base space (i.e. the plane or the sphere after a compactification procedure). In other words, for a given waveband defined for $\mathbf{k} \in \mathbb{R}^2$, the integration of the Berry curvature over $\mathbb{R}^2$ is not, in general, a Chern number, and even when it is, the bulk-edge correspondence is modified in a subtle way [63]. However, in any case, the monopole Chern numbers $\mathcal{C}$ are still well-defined as soon as a band crossing point is found. Put differently, unidirectional interface states are still well defined as topological waves, even though the interface separates two bulk domains that have ill-defined topology because of their ill-defined band Chern numbers. In that case, the Berry-Chern monopoles are the natural and sufficient objects to consider.

To sum up, the same monopole-spectral flow correspondence naturally captures, at the same time, the interface states in Chern insulators, valley Hall insulators, semi-metallic systems and continuous media, as well as the bulk current of Weyl semi-metals in a magnetic field.

### 2.4.3 A polariton-like toy model

The goal of this paragraph is two-fold: First, to illustrate how to construct a toy model of a wave coupled to a resonator that generates a topological spectral flow; and second, to point out that a spectral flow does not necessarily imply the existence of uni-directional modes.

Consider a non-dispersive wave of frequency $\omega = ck_x$ that travels through a resonator of frequency $\Omega$. If the wave travels freely in the media, (i.e. if it does not couple to the resonator), the resulting dispersion relation of the system is simply given by the superposition of the two branches $\omega_1 = ck_x$ and $\omega_2 = \Omega$. This obviously gives a linear band-crossing at $\Omega = ck_x$, but in a 1D parameter space, spanned by $k_x$ only. To get topologically guided modes out of this simple situation, one needs to embed this two-fold degeneracy point in a larger, i.e. three-dimensional, parameter space $(k_x, k_y, m(y))$. That is, one needs to couple the wave with the resonator to make new modes (in the spirit of a polariton [64–70]) but in such a way that it involves the wave number $k_y$ and an external or internal parameter $m$. Moreover, in order

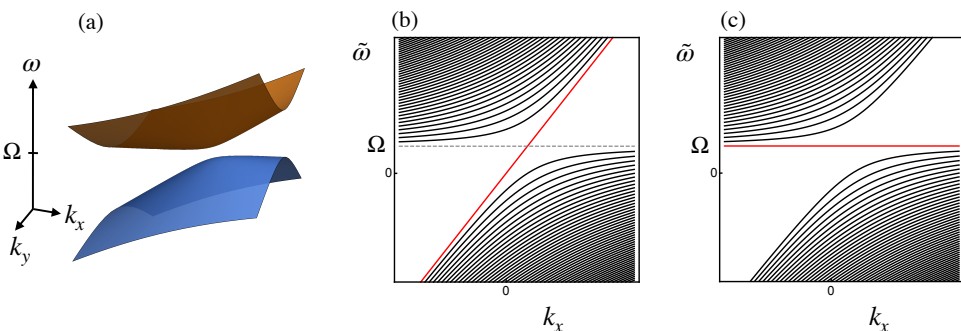

Figure 5: Frequency spectra of the polariton-like toy model for (a) the homogeneous (symbol Hamiltonian) problem, and (b) and (c) the inhomogeneous (operator Hamiltonian) problem, for (b) $m(y) \propto y$ and (c) $m(y) \propto -y$. The dashed line indicates the absence of a direct gap.

to embed the degeneracy point in the three-dimensional space, one sees that such a coupling $\gamma(k_y, m)$ must enter as a complex number, so that the Hamiltonian decomposes over the three Pauli matrices. One ends up with the following "polariton-like" toy model Hamiltonian

$$\begin{pmatrix} ck_x & \gamma^*(k_y, m) \\ \gamma(k_y, m) & \Omega \end{pmatrix} \begin{pmatrix} \varphi_1 \\ \varphi_2 \end{pmatrix} = \omega \begin{pmatrix} \varphi_1 \\ \varphi_2 \end{pmatrix}, \tag{59}$$

where $\varphi_1$ and $\varphi_2$ are complex numbers.

Its eigenfrequencies $\omega_\pm = \frac{ck_x + \Omega}{2} \pm \sqrt{\left(\frac{ck_x - \Omega}{2}\right)^2 + |2\gamma(k_y, m)|^2}$, are degenerated at the point $(k_x^0 = \frac{\Omega}{c}, k_y^0, m^0)$ with $\gamma(k_y^0, m^0) = 0$ by construction. Because this is a two-band crossing point, it generates a non-zero Berry-Chern monopole in $(k_x, k_y, m)$ space. If, moreover, $\gamma(k_y, m)$ varies linearly with $k_y$ and $m$ around the degeneracy point, so that the relation dispersion is linear around the band crossing point, then its topological charge is $|\mathcal{C}_\pm| = 1$. For the sake of simplicity, we choose $\gamma = k_y + im$, and plot the eigenfrequencies in figure 5. With this choice, this classical toy model looks like the 2D spin $S = 1/2$ massive Dirac Hamiltonian, except that the dispersion relation is tilted such that there is no direct gap. This however does not prevent the existence of a spectral flow which is guaranteed in the dual inhomogeneous operator problem defined after the substitutions $m \to m(y)$ and $k_y \to -i\partial_y$ provided $m(y)$ changes sign.[5] The value of the spectral flow $\Delta\mathcal{N}_\pm$ is given by $-\mathcal{C}_\pm$. The gain or the lost of a mode (say) in the band $+$, is then fixed by the sign of $\mathcal{C}_+$. This sign of the Chern numbers $\mathcal{C}_\pm$ is reversed when reversing the mass term $m \to -m$ in the model, and accordingly the spectral flow is reversed if $m(y)$ is a decreasing function of $y$ instead of an increasing one. As shown in figure 5 (b) and (c), this has a strong incidence on the physical behavior, since an unidirectional mode is obtained when $m(y)$ is an increasing function of $y$, while a non-propagating flat interface mode is obtained for a decreasing $m(y)$. Note that in both cases, there is not a direct gap. This is the cause of the absence of two counter propagating interface modes when swapping the sign of the mass term.

The general model addressed in the next section also exhibits non-chiral, non uni-directional spectral flows of non-zero energy modes, and spectral flows without a gap in the spectrum of the operator, in a natural and systematic way.

---

[5]We have assumed for simplicity that the celerity $c$ is a constant. Otherwise, the correspondence with the operator is slightly more involved.

# 3 Berry-Chern monopole - spectral flow correspondence beyond linear two-band crossings

## 3.1 $h.\hat{S}$ models

### 3.1.1 Motivations and generalities

In the first part of this manuscript, the monopole-spectral flow correspondence was discussed from the example of a spin 1/2 Hamiltonian, that is, for linear two-band crossings. Actually, the correspondence is neither restricted to two-fold band crossing points, nor to linear dispersion relations. In particular, a spectral flow of Landau levels was computed for a spin $S = 1$ generalization of Weyl semi-metals [27] and even for arbitrarily higher values of $S$ [19]. In the realm of classical physics, the spectral flow of oceanic and atmospheric equatorial waves obtained from the rotating shallow water model, that involves a three-band crossing point, was shown to have a topological interpretation from its symbol [15,71]. Remarkably, the spin 1 semi-metals under a magnetic field on the one hand, and the rotating shallow water model on the other hand, display both the exact same spectrum and carry the same Berry-Chern monopoles induce by three-fold degeneracy points. Although these two systems are physically radically distinct, they are in fact both described with the same spin $S = 1$ generalizations of the 3D Weyl semimetals and of the 2D massive Dirac particle examples, respectively. In that case, the Chern numbers take the values $\mathcal{C} = \{0, \pm 2\}$, and the spectral flow increases accordingly, so that the correspondence is satisfied. This spectral flow corresponds to a transfer of two Landau levels in the first case, and to the existence of two eastward propagating waves trapped along the equator, in the second one.

The goal of this section is to discuss the generalization of the monopole-spectral flow correspondence beyond both the spin 1/2 case, and linear dispersion relations. For that purpose, we will focus on models, that we shall dub $h.\hat{S}$ *models*, as their symbol Hamiltonians read

$$H_S = \boldsymbol{h}(\boldsymbol{\lambda}).\hat{\mathbf{S}}, \tag{60}$$

with $\boldsymbol{h} \in \mathbb{R}^3$. $\boldsymbol{\lambda} \in \mathbb{R}^3$ is a set of three *parameters,* including classical conjugate variables that one needs to quantize to obtain the operator Hamiltonian. $\hat{\mathbf{S}}$ is abusively called the spin operator: $\hbar$ has been removed from the usual definition, but its three components form an irreducible representation of the su(2) algebra embedded in su(N) $[\hat{S}_\alpha, \hat{S}_\beta] = \mathrm{i}\epsilon_{\alpha\beta\gamma}\hat{S}_\gamma$, where $\epsilon_{\alpha\beta\gamma}$ is the Levi-Civita symbol and $\{\alpha, \beta, \gamma\} = \{x, y, z\}$. It is worth stressing that the $h.\hat{S}$ models do not necessarily involve any actual spin. It turns out that such a structure also appears naturally in purely classical waves problems (an example is detailed in the appendix C). What only matters is this algebraic structure that is convenient to derive analytical results.

We choose $z$ as the quantization axis of $\hat{\mathbf{S}}$. Thus $\hat{S}_z$ is diagonal in the spin projection basis $|m_S\rangle$

$$\hat{S}_z |m_S\rangle = m_S |m_S\rangle, \tag{61}$$

with $m_S$ taking the $2S + 1$ integer or half integer values

$$m_S = -S, -S + 1, \ldots, S - 1, S. \tag{62}$$

In the following we will also make use of the spin ladder operators

$$\hat{S}_\pm = \hat{S}_x \pm \mathrm{i}\hat{S}_y \tag{63}$$

that satisfy

$$\hat{S}_\pm |m_S\rangle = \sqrt{S(S + 1) - m_S(m_S \pm 1)}|m_S \pm 1\rangle. \tag{64}$$

As we shall see, an elegant general expression of the Chern numbers of $h.\hat{S}$ models can be obtained analytically. After deriving this expression, we shall choose a specific $h.\hat{S}$ model, already introduced by Ezawa [19], for which we will not only directly obtain the Chern numbers but also investigate the operator Hamiltonian and its spectral flow. To do so, we will introduce a second index, in the spirit of the index of Dirac operators, and compute it to verify its equality with the Chern number, in agreement by the Atiyah-Singer theorem, which is the most advanced and abstract formulation of the monopole-spectral flow correspondence. Finally, we will construct the solutions of such a model and plot the different spectral flows.

### 3.1.2 Chern numbers of the $h.\hat{S}$ models

It is instructive to decompose the computation of the Chern numbers $\mathcal{C}_m$ for the Hamiltonian (60) into two steps, by focusing first on the linear case $\mathbf{h}(\boldsymbol{\lambda}) = \boldsymbol{\lambda}$ for an arbitrary spin $S$ before generalizing to arbitrary functions $\mathbf{h}(\boldsymbol{\lambda})$. This will allow us to distinguish two contributions to the Chern numbers.

**Revisiting Berry's original problem.** Let us start by setting $\mathbf{h}(\boldsymbol{\lambda}) = \boldsymbol{\lambda}$ in (60). The Hamiltonian

$$H_S = \boldsymbol{\lambda}.\hat{\mathbf{S}} \tag{65}$$

is then formally equal to that of a quantum spin coupled to a magnetic field $\boldsymbol{\lambda} = -\mu_B \mathbf{B}$ with $\mu_B$ the Bohr magneton. This model was discussed by Michael Berry in its seminal paper [45] to introduce the geometrical phase accumulated by a quantum state $|\psi_{m_S}\rangle$ after varying adiabatically the magnetic field along a loop in space. The adiabaticity invoked here imposes the eigenstates $|\psi_{m_S}\rangle$ of $H_S$ to remain eigenstates during the evolution while $\mathbf{B}$ is varied. In our formal model, $\boldsymbol{\lambda}$ is not physically varied in time; it is just a parameter of the Hamiltonian and of its eigenstates.

To compute the geometric phase of the eigenstates of $H_S$, Berry computed the 2-form Berry curvature. The Berry phase is then obtained by integrating this curvature along a close path, while the Chern number is obtained after integrating the Berry curvature over a close surface. Berry's computation of the curvature is based on the expression (41) of the coefficients $F_{ij}^{(m_S)}(\boldsymbol{\lambda})$, so that the curvature reads

$$\mathcal{F}_{m_S}(\boldsymbol{\lambda}) = i \sum_{p \neq m_S} \frac{\langle \psi_{m_S}(\boldsymbol{\lambda})| \partial_{\lambda_i} H_S |\psi_p(\boldsymbol{\lambda})\rangle \langle \psi_p(\boldsymbol{\lambda})| \partial_{\lambda_j} H_S |\psi_{m_S}(\boldsymbol{\lambda})\rangle}{(E_p(\boldsymbol{\lambda}) - E_{m_S}(\boldsymbol{\lambda}))^2} d\lambda_i \wedge d\lambda_j. \tag{66}$$

This expression of the Berry curvature can be cumbersome to manipulate in cartesian coordinates, especially when one aims at integrating it over the sphere to get the Chern numbers. It is thus much more judicious to work in spherical coordinates, and the symbol Hamiltonian then becomes

$$H_S = |\boldsymbol{\lambda}| \, \mathbf{n}(\theta, \phi) \cdot \hat{\mathbf{S}}, \tag{67}$$

where the normalized vector $\boldsymbol{\lambda}/|\boldsymbol{\lambda}| \equiv \mathbf{n}(\theta, \phi) = (\sin\theta\cos\phi, \sin\theta\sin\phi, \cos\theta)$ gives the orientation of $\boldsymbol{\lambda}$ in $\mathbb{R}^3$, with $\theta$ and $\phi$ the usual polar and azimuthal angles in the laboratory frame respectively. Our three abstract parameters $\{\lambda_i\}$ in (66) must now be understood as the new coordinates $\{|\boldsymbol{\lambda}|, \theta, \phi\}$. One now needs to express explicitly $H_S$ with those new variables, so that one can effectively differentiate it according to (66).

Note that there is an ambiguity here about what $|\psi_{m_S}\rangle$ means. Indeed, $m_S$ refers to the projection of the spin onto a quantization axis. Usually, in problems involving a Zeeman coupling,

the magnetic field's orientation is fixed, and canonically provides the axis of quantization. Here the situation is more subtle, since the orientation of the magnetic field, given by $\boldsymbol{n}(\theta,\phi)$, is not fixed, so one must precise which axis we refer to when writing $|\psi_{m_S}\rangle$. Let us thus fix the quantization axis, called $z$, in the laboratory frame. Then one writes $|m_S,\mathbf{n}\rangle$ to refer to an eigenstate of $H_S$, and $|m,\mathbf{e}_z\rangle$ to refer to an eigenstate of $\hat{S}_z$ i.e.

$$\hat{S}_z\,|m_S,\mathbf{e}_z\rangle = m_S\,|m_S,\mathbf{e}_z\rangle\,, \tag{68}$$

$$H_S\,|m_S,\mathbf{n}\rangle = |\boldsymbol{\lambda}|\,m_S\,|m_S,\mathbf{n}\rangle\,, \tag{69}$$

where $\hat{S}_z = \mathbf{e}_z\cdot\hat{\mathbf{S}}$. The eigenstates that intervene in the Berry curvature (66) must therefore be understood as $|\psi_{m_S}\rangle \equiv |m_S,\mathbf{n}\rangle$.

The operator $\hat{S}_z$ is related to the Hamiltonian $\mathbf{n}\cdot\hat{\mathbf{S}}$ by a rotation, represented by a unitary operator $U(\theta,\phi)$ that depends on the longitudinal and azimutal angles as

$$\mathbf{n}(\theta,\phi)\cdot\hat{\mathbf{S}} \equiv U(\theta,\phi)\,\mathbf{e}_z\cdot\hat{\mathbf{S}}\,U^\dagger(\theta,\phi) = U(\theta,\phi)\,\hat{S}_z\,U^\dagger(\theta,\phi)\,. \tag{70}$$

This relates the eigenstates of the "rotated" Hamiltonian $H(\theta,\phi)$ to those of $\hat{S}_z$ as

$$|m_S,\mathbf{n}\rangle \equiv U(\theta,\phi)\,|m_S,\mathbf{e}_z\rangle\,. \tag{71}$$

The rotation operator $U(\theta,\phi)$ is not uniquely defined. For instance, one can choose the representation

$$U(\theta,\phi) = \mathrm{e}^{-\mathrm{i}\phi\hat{S}_z}\mathrm{e}^{-\mathrm{i}\theta\hat{S}_y}\mathrm{e}^{\mathrm{i}\phi\hat{S}_z}\,. \tag{72}$$

The Berry curvature (66) can now be computed. It implies terms of the form $\langle m,\mathbf{n}|\,\partial_\theta H_S\,|p,\mathbf{n}\rangle$, $\langle m,\mathbf{n}|\,\partial_\phi H_S\,|p,\mathbf{n}\rangle$ and $\langle m,\mathbf{n}|\,\partial_{|\boldsymbol{\lambda}|}H_S\,|p,\mathbf{n}\rangle$, but one can notice that all the terms involving a derivative with respect to the radius $|\boldsymbol{\lambda}|$ vanish, because they read $\langle m,\mathbf{e}_z|\,p\,|p,\mathbf{e}_z\rangle$ with $m\neq p$. We are thus left with a pure angle dependence only. Recalling the expression of a rotation of $\hat{S}_z$ by an angle $\theta$ around the $y$ axis

$$\mathrm{e}^{-\mathrm{i}\theta\hat{S}_y}\hat{S}_z\mathrm{e}^{\mathrm{i}\theta\hat{S}_y} = \cos\theta\hat{S}_z + \sin\theta\hat{S}_x \tag{73}$$

one finds

$$\langle m_S,\mathbf{n}|\,\partial_\theta H_S\,|p,\mathbf{n}\rangle = |\boldsymbol{\lambda}|\,\langle m_S,\mathbf{e}_z|\hat{S}_x\,|p,\mathbf{e}_z\rangle \tag{74}$$

$$\langle p,\mathbf{n}|\,\partial_\phi H_S\,|m_S,\mathbf{n}\rangle = |\boldsymbol{\lambda}|\,\sin\theta\,\langle p,\mathbf{e}_z|\hat{S}_y\,|m_S,\mathbf{e}_z\rangle\,. \tag{75}$$

These quantities can be computed easily by using the spin ladder operators through the substitution $\hat{S}_x = \frac{1}{2}(\hat{S}_+ + \hat{S}_-)$ and $\hat{S}_y = \frac{1}{2\mathrm{i}}(\hat{S}_+ - \hat{S}_-)$, and one ends up with the important result

$$\mathcal{F}_{m_S} = -m_S\,\sin\theta\,\mathrm{d}\theta\wedge\mathrm{d}\phi\,, \tag{76}$$

that was first found by Berry [45]. In this expression, one recognizes (see appendix B) the surface form $4\pi\Omega_{S^2} = \sin\theta\mathrm{d}\theta\wedge\mathrm{d}\phi$ of the sphere $S^2$ in spherical coordinates that satisfies $\int_{S^2}\Omega_{S^2} = 1$. The Chern numbers are then readily obtained by integrating (76) over $S^2$ so that

$$\mathcal{C}_{m_S} = -2m_S\,. \tag{77}$$

This result was originally derived in a different way by Avron, Sadun, Segert and Simon [72]. In the case of a spin $1/2$, one recovers the result $\mathcal{C}_\pm = \mp 1$ for the two eigenstates $m_S = \pm 1/2$ as derived previously. Higher values of the Chern number can then be reached for higher spins.

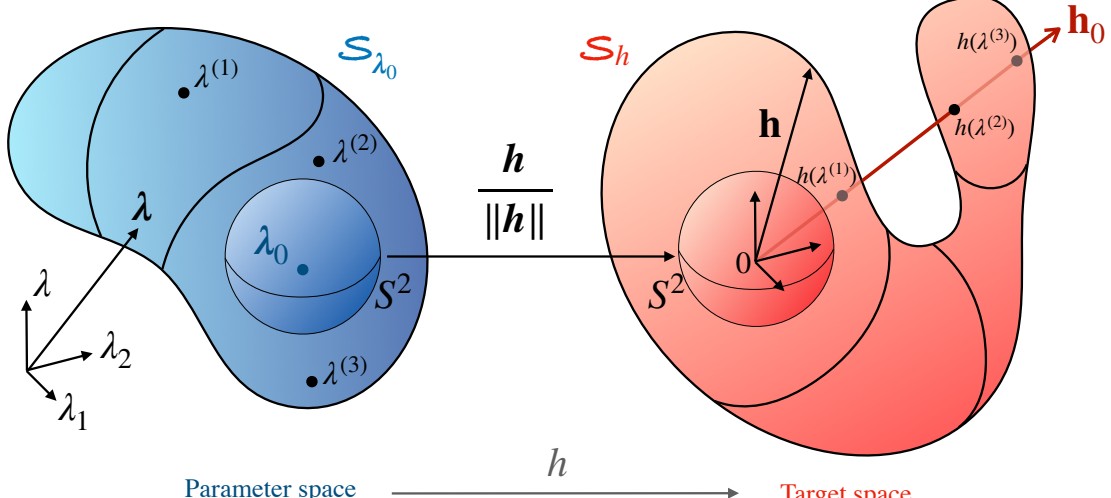

Figure 6: Sketch of the Hamiltonian map $h$ between the parameter space and the target space. The degree of $h$ is defined when considering the image $\mathcal{S}_h$ of a surface $\mathcal{S}_{\lambda_0}$ that encloses the degeneracy point $\lambda_0$. The points $\lambda^{(1)}$, $\lambda^{(2)}$ and $\lambda^{(3)}$ are examples of pre-imagines that enter the formula (79) of the degree.

**The *Hamiltonian map* and its degree.** Let us now reintroduce the full $\mathbf{h}(\lambda)$ dependence in the problem. We shall refer to $\lambda \to \mathbf{h}(\lambda)$ as the *Hamiltonian map* $h$ between the parameter space of the $\lambda$'s and a target space $\mathbb{R}^3$ spanned by $\mathbf{h}$. In others words, the image of $\lambda$ by the Hamiltonian map $h$ is represented by a vector $\mathbf{h} \in \mathbb{R}^3$ (see figure 6). As motivated in section 2.3.3, one can view the degeneracy points $\lambda_0$ as defects in parameter space. Those defects can be characterized by the way the Hamiltonian map transforms a surface $\mathcal{S}_{\lambda_0}$ enclosing $\lambda_0$, into a surface $\mathcal{S}_h$ in target space, that is by considering [73]

$$\lambda \in \mathcal{S}_{\lambda_0} \subset \mathbb{R}^3 \to \mathbf{h}(\lambda) \in \mathcal{S}_h \subset \mathbb{R}^3 \,. \tag{78}$$

Since those two surfaces are oriented manifold and have the same dimension, the maps $h$ between those spaces are classified according to their *degree* that tells whether $\mathcal{S}_h$ is a closed surface surrounding the origin $\mathbf{h} = 0$ and how many times it encloses it [73]. It reads

$$\deg h \equiv \sum_{\lambda^{(i)}} \operatorname{sgn} \det \begin{pmatrix} \partial_{\lambda_1} h_x & \partial_{\lambda_1} h_y & \partial_{\lambda_1} h_z \\ \partial_{\lambda_2} h_x & \partial_{\lambda_2} h_y & \partial_{\lambda_2} h_z \\ \partial_{\lambda_3} h_x & \partial_{\lambda_3} h_y & \partial_{\lambda_3} h_z \end{pmatrix} \Bigg|_{\mathbf{h}_0} , \tag{79}$$

where the matrix whose determinant is computed is the Jacobian matrix between the parameter space and the target space. The points $\lambda^{(i)}$, which the sum runs over, are called the pre-images of $h$, and are associated to a given arbitrary direction $\mathbf{n}_0 \equiv \mathbf{h}_0/|\mathbf{h}_0|$, as they satisfy $\mathbf{h}(\lambda^{(i)})/|\mathbf{h}(\lambda^{(i)})| = \mathbf{n}_0$. So in other words, the pre-images $\lambda^{(i)}$ are all the points of parameter space whose image $\mathbf{h}(\lambda^{(i)})$ points in a fixed direction $\mathbf{n}_0$ (see figure 6). The value of the degree does not depend on the choice of that direction, provided the determinant is not singular (i.e. nonzero) for that choice.

The degree is obviously an integer number. This is a topological index in that it is invariant under continuous changes of $h$. Two maps $h_1$ and $h_2$ having the same degree can thus be continuously deformed one into each others; they are said to be homotopic. This homotopy invariance makes free the choice of surfaces $\mathcal{S}_{\lambda_0}$ that surrounds the degeneracy point $\lambda_0$ so that one can choose it as being the unit sphere (see figure 6). The degree actually classifies

the maps from spheres to spheres

$$\boldsymbol{\lambda} \in S^2 \to \mathbf{h}/|\mathbf{h}| \in S^2, \tag{80}$$

according to their homotopy properties. They are the elements of the homotopy group $\pi_2(S^2) = \mathbb{Z}$ (and more generally that of $\pi_n(S^n) = \mathbb{Z}$). As for the maps $h$ we deal with, the degree is known to have an elegant geometrical interpretation: it counts how many times the unit vector $\mathbf{h}/|\mathbf{h}|$ in (80) wraps the unit sphere (see appendix B for more details).

To see explicitly how the degree alters the value of the Chern number, one must interpret the result of the previous section as if the reasoning was performed in the target space. Namely, the expression of the Berry curvature (76) must be understood in the space of $\mathbf{h}$ as if the underlying $\boldsymbol{\lambda}$ dependence was ignored. However, we want to express the Chern number as an integral of the Berry curvature over a surface (a sphere) that surrounds the degeneracy point in parameter space $\Lambda$ as

$$\mathcal{C}_{m_S} = \frac{1}{2\pi} \int_{S^2} \mathcal{F}_{m_S} \tag{81}$$

$$= -2m_S \int_{S^2} h^\star \Omega_{S^2}, \tag{82}$$

where $h^\star \Omega_{S^2}$ denotes the *pull-back* of $\Omega_{S^2}$ by $h$; this is a procedure to formally define a differential form in parameter space when it is already defined in target space (see appendix B). Actually this formal manipulation simplifies easily thanks to the Brouwer theorem, that relates the integrals of a differential form to that of its "pulled-back" as

$$\mathcal{C}_{m_S} = -2m_S \deg h \int_{S^2} \Omega_{S^2}, \tag{83}$$

where the integral acts now in target space and where $\deg h$ is precisely the degree of $h$. Again, the integral that remains is 1 by definition, which leads to the final expression of the topological charges of the Berry-Chern monopoles

$$\mathcal{C}_{m_S} = -2m_S \deg h. \tag{84}$$

This is the general and practical expression of the Chern-Berry monopole for $h.\hat{S}$ models [74, 75].

One can easily check that one recovers $\mathcal{C}_\pm = \mp 1$ for the $2 \times 2$ Hamiltonians ($m_S = \pm 1/2$), since the degree (79) in that case is 1. It is worth noticing that the formula (84) clearly reveals the two concurrent mechanisms that give the Chern numbers their values : the order of the degeneracy (property of $\hat{\mathbf{S}}$) and the wrapping of the Hamiltonian map (property of $\mathbf{h}(\boldsymbol{\lambda})$).

## 3.2 A specific $h.\hat{S}$ model

In this section, we propose a specific $h.\hat{S}$ model to decorticate the correspondence between the spectral flow of an operator Hamiltonian and the Chern numbers of the fiber bundles associated to the eigenstates of the symbol Hamiltonian. This model generalizes what was discussed in the first part of the manuscript to (1) arbitrary integer and half-integer spins $S \geqslant 1/2$ and (2) higher order ladder operators $\hat{a} \to \hat{a}^d$ with $d$ a positive integer. The operator Hamiltonian we consider is a composition of spin ladder operators $\hat{S}_\pm$ and bosonic ladder operators $\hat{a}$ and $\hat{a}^\dagger$ that reads

$$\hat{\mathcal{H}}_{\mathrm{op}} = \left( (\hat{a}^\dagger)^d \, \hat{S}_+ + \hat{a}^d \, \hat{S}_- \right) + \lambda \hat{S}_z. \tag{85}$$

Up to a factor 2, this new $\hat{\mathcal{H}}_{\text{op}}$ coincides with the previously studied operator Hamiltonian (Eqs (7) and (16)) when $S = 1/2$ and $d = 1$. It acts on a Hilbert space that decomposes as the product of a Fock space $\mathcal{F}$, where the number states $|n\rangle$ live, and a spin space $\mathcal{S}$ spanned by the spin projections $|m_S\rangle$. A vector of this space $\mathcal{F} \otimes \mathcal{S}$ thus reads $|n\rangle \otimes |m_S\rangle \equiv |n, m_S\rangle$. The action of $\hat{\mathcal{H}}_{\text{op}}$ on those states is then to shift the different components $n$ and $m_S$; it thus somehow describes a dynamics on a lattice of coordinates $(n, m_S)$.

As we shall see, this model exhibits very rich spectral flows, depending on $S$ and $d$. In the limit $|\lambda| \gg 1$, which is precisely when the spectral flow modes join a branch, the term $\lambda \hat{S}_z$ is dominant, and one expects therefore the eigenstates $|\Psi\rangle$ of $\hat{\mathcal{H}}_{\text{op}}$ to resemble the spin states $|m_s\rangle$, because the typical energy between two consecutive number states $|n\rangle$ and $|n+1\rangle$ in the same branch must tend to zero. There is then a clear separation of energy (or time) scales, such that the spin contribution $\lambda \hat{S}_z$ (that one can associate to a fast motion), is dominant over the $\hat{a}^d \hat{S}_-$ and $\hat{a}^\dagger \hat{S}_+$ terms that imply annihilation and creation operators (and that one can thus associate to a slow motion). With this in mind, it is tempting, in order to capture the spectral flow occurring for $|\lambda| \gg 1$, to approximate the system by an "adiabatic" model, by substituting the operator $\hat{a}^d$ by its symbol. Such a transformation is *a priori* not straightforward, since the symbol of a product of operators is in general not equal to the product of the operators of each symbols [41, 42]. It turns out that this is however the case in our model, and we have the simple correspondences

$$(\lambda_1 + i\lambda_2)^d \quad \leftrightarrow \quad \hat{a}^d, \tag{86}$$

$$(\lambda_1 - i\lambda_2)^d \quad \leftrightarrow \quad (\hat{a}^\dagger)^d, \tag{87}$$

where $\lambda_1$ and $\lambda_2$ are canonical conjugate variables in phase space. This mapping is a consequence of a result derived long ago by Mac Coy [76], and that is re-derived by induction in the appendix A.

One thus obtains the symbol Hamiltonian of the expected form $H_S = \boldsymbol{h}(\boldsymbol{\lambda}) . \hat{\mathbf{S}}$ as announced in (60), with

$$h_x(\boldsymbol{\lambda}) = 2 \operatorname{Re}(\lambda_1 - i\lambda_2)^d \tag{88a}$$

$$h_y(\boldsymbol{\lambda}) = -2 \operatorname{Im}(\lambda_1 - i\lambda_2)^d \tag{88b}$$

$$h_z(\boldsymbol{\lambda}) = \lambda. \tag{88c}$$

The Hilbert space of the symbol Hamiltonian is reduced, compared to that of $\hat{\mathcal{H}}_{\text{op}}$, since only the $2S+1$ spin (fast) degrees of freedom are kept, but is parametrized by the slow degrees of freedom that are accounted by the commuting (classical) variables. In that picture, the space of fast motion is a fiber bundle over the phase space of slow variables [40, 71].

In the next section, we introduce the analytical index of the operator Hamiltonian (85), relate it to the spectral flow, compute it and compare its values to that of the Chern numbers of $H_S$ with (88).

## 3.3 Analytical and Topological indices, a verification of the Atiyah-Singer theorem

### 3.3.1 Chern numbers of the specific $h.\hat{S}$ model

The expression (84) of the Chern numbers is quite general. In this section we compute it for the specific symbol Hamiltonian given by Eqs (88). One thus needs to compute the degree of

such a Hamiltonian map $h$, which requires the determinant of the Jacobian matrix (79) :

$$\det\begin{pmatrix} \partial_{\lambda_1} h_x & \partial_{\lambda_1} h_y & 0 \\ \partial_{\lambda_2} h_x & \partial_{\lambda_2} h_y & 0 \\ 0 & 0 & 1 \end{pmatrix} = \frac{\partial h_x}{\partial \lambda_1}\frac{\partial h_y}{\partial \lambda_2} - \frac{\partial h_x}{\partial \lambda_2}\frac{\partial h_y}{\partial \lambda_1}. \tag{89}$$

We have

$$(\lambda_1 - \mathrm{i}\lambda_2)^d = \sum_{\alpha=0}^{d}\binom{d}{\alpha}\lambda_1^{d-\alpha}(-\mathrm{i}\lambda_2)^\alpha, \tag{90}$$

that we split into even $\alpha = 2k$ and odd $\alpha = 2k+1$ contributions to get the real and imaginary parts, so that

$$h_x = 2\sum_{\substack{k\geqslant 0 \\ 2k\leqslant d}}\binom{d}{2k}(-1)^k\lambda_1^{d-2k}\lambda_2^{2k}, \tag{91}$$

$$h_y = 2\sum_{\substack{k\geqslant 0 \\ 2k+1\leqslant d}}\binom{d}{2k+1}(-1)^k\lambda_1^{d-2k-1}\lambda_2^{2k+1}. \tag{92}$$

To compute the determinant (89), one then needs to take the derivatives of these terms. One has

$$\frac{\partial h_x}{\partial \lambda_2} = 2\sum_{\substack{k\geqslant 1 \\ 2k\leqslant d}}(-1)^k\binom{d}{2k}(2k)\lambda_1^{d-2k}\lambda_2^{2k-1}, \tag{93}$$

$$\frac{\partial h_y}{\partial \lambda_1} = 2\sum_{\substack{k\geqslant 0 \\ 2k+1\leqslant d-1}}(-1)^k\binom{d}{2k+1}(d-2k-1)\lambda_1^{d-2k-2}\lambda_2^{2k+1}. \tag{94}$$

By setting $k' = k+1$, one can rewrite the second equation as

$$\frac{\partial h_y}{\partial \lambda_1} = -2\sum_{\substack{k'\geqslant 1 \\ 2k'\leqslant d}}(-1)^{k'}\binom{d}{2k'-1}(d-(2k'-1))\lambda_1^{d-2k'}\lambda_2^{2k'-1}, \tag{95}$$

and notice that

$$\binom{d}{2k'-1}(d-(2k'-1)) = \binom{d}{2k}2k, \tag{96}$$

so that

$$\frac{\partial h_y}{\partial \lambda_1} = -\frac{\partial h_x}{\partial \lambda_2}. \tag{97}$$

Repeating this algebraic gymnastic for the cross derivatives terms yields now

$$\frac{\partial h_x}{\partial \lambda_1} = \frac{\partial h_y}{\partial \lambda_2} \tag{98}$$

and thus the determinant (89) satisfies

$$\det\begin{pmatrix} \partial_{\lambda_1} h_x & \partial_{\lambda_1} h_y & 0 \\ \partial_{\lambda_2} h_x & \partial_{\lambda_2} h_y & 0 \\ 0 & 0 & 1 \end{pmatrix} = \left(\frac{\partial h_x}{\partial \lambda_1}\right)^2 + \left(\frac{\partial h_y}{\partial \lambda_1}\right)^2 > 0. \tag{99}$$

Therefore, each pre-image $\lambda^{(i)}$ contributes as $+1$ to to the degree (79), that is

$$\deg h = \sum_{\lambda^{(i)}} 1 \,. \tag{100}$$

The computation of the degree now simply consists in counting the number of pre-images. To complete this calculation, one must choose a direction $\mathbf{n}_0$ in the target space. The choice $\mathbf{n}_0 = \mathbf{e}_z$ is not an option, since it would imply that the expression (99) of the determinant must be evaluated at $\lambda_1 = \lambda_2 = 0$ (as soon as $d > 1$), which yields a vanishing determinant. Instead, one can choose any direction in the plane $(\mathbf{e}_x, \mathbf{e}_y)$, such that, e.g. $\mathbf{n}_0 = \mathbf{e}_y$. Using the polar coordinates $(\lambda_1 - i\lambda_2) \equiv z = |z|e^{i\theta}$, the number of pre-imagines to be determined is the number of values $\theta_p$ that satisfy

$$h_x = 2|z|^d \cos(\theta_p d) = 0 \tag{101}$$

$$h_y = 2|z|^d \sin(\theta_p d) > 0 \,. \tag{102}$$

This involves

$$\theta_p = \frac{\pi}{2d} + 2\pi \frac{p}{d} \,, \tag{103}$$

with $p$ an integer. It follows that $\theta$ can take $d$ different values on the circle to satisfy those conditions, this means that there are $d$ pre-images.

Finally, the degree of the hamiltonian map is therefore $\deg h = d$ and the values of the Chern numbers of the symbol Hamiltonian for the model (88) are

$$\mathcal{C}_{m_S} = -2m_S d \,. \tag{104}$$

### 3.3.2  Spectral flow as an analytical index of $\hat{\mathcal{H}}_{\mathbf{op}}$

We would like to introduce an index [37, 77–79] that accounts for the spectral flow of the operator Hamiltonian $\hat{\mathcal{H}}_{\mathrm{op}}$ that we have introduced in equation (85). We expect then that this index to be related to the the first Chern numbers (104), according to index theorems [20, 26, 37, 38].

In the first part, we introduced the notion of spectral flow from the canonical example of a two-band model ($\hat{\mathcal{H}}_{\mathrm{op}}$ with $S = 1/2$ and $d = 1$) in a handwavy way, as the number of energy levels that transit from one branch to another when a parameter $\lambda$ is continuously varied.[6] For this same model, the spectral flow can be defined more formally by a *spectral index* that counts the net number of energy levels that move upward in the vicinity of $\tilde{E} = 0$ and $\lambda = 0$, when sweeping $\lambda$. The existence of such an index is meaningful provided the spectrum $\tilde{E}$ of $\hat{\mathcal{H}}_{\mathrm{op}}$ is discrete within that window. This discreteness is guaranteed for this model since the symbol Hamiltonian has the particular property of being an *elliptic* operator; its determinant is non-zero except at the degeneracy point [26, 71]. This spectral index is however not obviously suitable beyond $S > 1/2$, since more than two bands are involved. Besides, as we shall see with the specific $\hat{\mathcal{H}}_{\mathrm{op}}$ we introduced in (85), other spectral flows can exist between the different branches, without crossing the spectral gap around $\tilde{E} = 0$ and $\lambda = 0$. We thus need to follow a different strategy.

Alternatively, we could also define an index that counts the number of energy modes that join or leave each branch when $|\lambda| \to \infty$, without specifying an energy window around the degeneracy point of $H_S$. For that purpose, one can get some insights from another quite standard procedure to define a spectral flow, although more abstract, that was developed for Dirac

---

[6]See for instance page 99 of [39] for a much more rigorous and abstract definition.

Operators [26, 78, 80]. Dirac operators are first order differential operators whose square is a Laplacian. They decompose over $2N + 1$ $\gamma$ matrices which satisfy the Clifford algebra $\gamma^j \gamma^k + \gamma^k \gamma^j = 2\delta^{jk} \mathbb{I}_{2N}$. This algebraic structure guarantees a chiral symmetry; namely, there necessarily exists a unitary operator $\Gamma$ that can be written as $\Gamma = \mathrm{diag}(\mathbb{I}_N, -\mathbb{I}_N)$ and that anti-commutes with the Dirac Hamiltonian. It follows that the Dirac Hamiltonian can always take an off-diagonal form as

$$\mathcal{H}_{Dirac} = \begin{pmatrix} 0 & D^\dagger \\ D & 0 \end{pmatrix}. \tag{105}$$

For instance, in the particular case $N = 1$, the $\gamma$ matrices reduces to the Pauli matrices and $D = \partial_x + A(x)$ with $A(x)$ a real function [26]. The chiral structure of the Dirac Hamiltonian splits the Hilbert space into two chiral subspaces $E_- \oplus E_+$ defined from the spectral projectors $\mathcal{P}_+ \equiv \frac{1}{2}(\mathbb{I}_{2N} + \Gamma) = \mathrm{diag}(\mathbb{I}_N, 0_N)$ and $\mathcal{P}_- \equiv \frac{1}{2}(\mathbb{I}_{2N} - \Gamma) = \mathrm{diag}(0_N, \mathbb{I}_N)$. A state living in $E_\pm$ is thus an eigenstate of $\Gamma$ with eigenvalue $\pm 1$, called the chirality. The two chiral subspaces are related to each other as

$$E_- \underset{\hat{D}}{\overset{\hat{D}^\dagger}{\rightleftarrows}} E_+, \tag{106}$$

where

$$\hat{D} = \begin{pmatrix} 0 & 0 \\ D & 0 \end{pmatrix}, \qquad \hat{D}^\dagger = \begin{pmatrix} 0 & D^\dagger \\ 0 & 0 \end{pmatrix}. \tag{107}$$

Structures such as (106), with nilpotent operators $\hat{D}$, are referred to as *complexes* [26]. They are associated to an *index* defined as [26, 78, 80]

$$\mathrm{ind}\,\hat{D} \equiv \dim \mathrm{Ker}\,\hat{D} - \dim \mathrm{Ker}\,\hat{D}^\dagger, \tag{108}$$

which is well-defined when the right-hand side member is finite. In that case, the index is obviously an integer number, and is known to be robust under homotopic deformations. In other words, multiplying $\hat{D}$ by a nonzero number or by an operator which is connected to Identity (such as a unitary) does not change the value of the index.

   As we shall see in a moment, $\mathrm{ind}\,\hat{D}$ yields an important information about the spectrum of the $\mathcal{H}$. Indeed, this index counts the unbalance of zero energy modes of opposite chiralities. The occurrence of such so-called zero modes is a fundamental property of fermions coupled to gauge fields. Their investigation for Dirac operators has stimulated tremendous efforts in high and low energy physics and in mathematics. Importantly for our purpose, the index of Dirac operators is also related to the spectral flow of zero-modes. It is then tempting to apply this machinery to $\hat{\mathcal{H}}_{\mathrm{op}}$ for $S = 1/2$ and $d = 1$, that is when the Hamiltonian operator $\hat{\mathcal{H}}_{\mathrm{op}}$ reduces to that of 3D Weyl semimetals in a magnetic field (16) as well as that of 2D Dirac fermions with an anisotropic mass (7). Indeed, in that case, and when we take $\lambda = 0$, $\hat{\mathcal{H}}_{\mathrm{op}}$ has a similar form as (105) with $D = \hat{a}$, $D^\dagger = \hat{a}^\dagger$, and the index $\mathrm{ind}\,\hat{D}$ can easily be found to be 1 by a straightforward calculation, in agreement with the spectral flow previously computed. However, this procedure seems to work only when $\lambda = 0$, since the chiral structure is lost in the presence of $\lambda$. Moreover, and more importantly, $\hat{\mathcal{H}}_{\mathrm{op}}$ does not satisfy anyway such a chiral symmetry beyond $S = 1/2$, meaning that $\hat{\mathcal{H}}_{\mathrm{op}}^2$ is not a Laplacian, even when $\lambda = 0$. In other words, $\hat{\mathcal{H}}_{\mathrm{op}}$ is not a Dirac operator.

   One can nevertheless circumvent this difficulty and generalize the previous approach by writing the operator Hamiltonian as

$$\hat{\mathcal{H}}_{\mathrm{op}} = \mathcal{D} + \mathcal{D}^\dagger + \lambda \hat{S}_z, \tag{109}$$

with

$$\mathcal{D} \equiv \hat{a}^d \hat{S}_-. \qquad \mathcal{D}^\dagger = (\hat{a}^\dagger)^d \hat{S}_+ \,, \tag{110}$$

that allows us to decompose the Hilbert space as

$$E_{-S} \underset{\mathcal{D}}{\overset{\mathcal{D}^\dagger}{\rightleftarrows}} \dots \underset{\mathcal{D}}{\overset{\mathcal{D}^\dagger}{\rightleftarrows}} E_{m_S-1} \underset{\mathcal{D}}{\overset{\mathcal{D}^\dagger}{\rightleftarrows}} E_{m_S} \underset{\mathcal{D}}{\overset{\mathcal{D}^\dagger}{\rightleftarrows}} E_{m_S+1} \cdot \underset{\mathcal{D}}{\overset{\mathcal{D}^\dagger}{\rightleftarrows}} \dots \cdot \underset{\mathcal{D}}{\overset{\mathcal{D}^\dagger}{\rightleftarrows}} E_S \,, \tag{111}$$

where the subspaces $E_{m_S}$ are spanned by states $|\Psi\rangle = |p, m_S\rangle$ with arbitrary $p \in \mathbb{N}$ and fixed $m_S$. Actually, one can also associate an index to such a structure that counts the spectral flow of $\hat{\mathcal{H}}_{\mathrm{op}}(\lambda)$.

Indeed, the spectral flow $\Delta\mathcal{N}_{m_S}$ measures the unbalance of number states $p$ inside a branch $m_S$, when $\lambda$ is swept from $-\infty$ to $+\infty$. One can use $\mathcal{D}$ to introduce an index that measures this unbalance, by noticing that $\hat{\mathcal{H}}_{\mathrm{op}}$ is invariant under the joint transformations $\boldsymbol{\lambda \to -\lambda}$ and $\hat{S}_z \to -\hat{S}_z$. In other words, flipping $\lambda$ has the same effect as flipping $\hat{S}_z$. Since this transformation amounts to perform $m_S \to -m_S$, then evaluating the spectral flow in a branch $m_S$ thus amounts to counting what is left after pairing the states $|p, m_S\rangle$ with $|p', -m_S\rangle$. Those states respectively live in $E_{m_S}$ and $E_{-m_S}$ spaces, that are related to each other by $\mathcal{D}^{2m_S}$, as sketched in Fig. 7, and we have

$$\mathcal{D}^{2m_S} |p, +m_S\rangle = \prod_{j=1}^{d} \sqrt{p + 2m_S(j-1)} \, |p - 2m_S d, -m_S\rangle \tag{112}$$

$$(\mathcal{D}^\dagger)^{2m_S} |p', -m_S\rangle = \prod_{j=1}^{d} \sqrt{p' + 2m_S j} \, |p' + 2m_S d, m_S\rangle \,. \tag{113}$$

Actually, the prefactors in Eqs. (112) and (113) are unimportant. What is important is the asymmetry of the action of $\mathcal{D}^{2m_S}$ and $(\mathcal{D}^\dagger)^{2m_S}$. Indeed, Eq. (113) indicates that any mode $p'$ in the branch $m_S$ for $\lambda < 0$ can be paired with a mode $p = p' + 2m_S d$ in the same branch for $\lambda > 0$, because $p$ cannot be negative. In other words, $(\mathcal{D}^\dagger)^{2m_S}$ is injective, and we have

$$\dim \mathrm{Ker}(\mathcal{D}^\dagger)^{2m_S} = 0 \,. \tag{114}$$

In contrast, $\mathcal{D}^{2m_S}$ associates a state $|p', -m_S\rangle \in E_{-m_S}$ for each state $|p, m_S\rangle \in E_{m_S}$ with $p' = p - 2m_S d$, but in a non-injective way, since

$$|p - 2m_S d, -m_S\rangle = 0 \qquad \mathrm{for} \quad p = 0, \dots, 2m_S d - 1. \tag{115}$$

It follows that the number of modes at $\lambda > 0$ in the branch $m_S$ that do not have a pairing partner in the same branch at $\lambda < 0$ through the action of $\mathcal{D}^{2m_S}$, is given by

$$\dim \mathrm{Ker}\mathcal{D}^{2m_S} = 2m_S d \,. \tag{116}$$

The figure 7 summarizes this peculiar structure induced by $\mathcal{D}$ being *almost* invertible. The unbalance $\Delta\mathcal{N}_{m_S}$ of modes in a branch $m_S$ between $\lambda > 0$ and $\lambda < 0$ is finally given by the net number of un-paired modes between $E_{m_S}$ and $E_{-m_S}$ as

$$\Delta\mathcal{N}_{m_S} = \dim \mathrm{Ker}\mathcal{D}^{2m_S} - \dim \mathrm{Ker}(\mathcal{D}^\dagger)^{2m_S} \,. \tag{117}$$

The expression (117) is similar to that of the index of Dirac operators introduced above. It is actually an index more generally associated to a class of operators called *Fredolhm* operators [20, 26], which are invertible operators modulo a compact operator, and whose symbol is

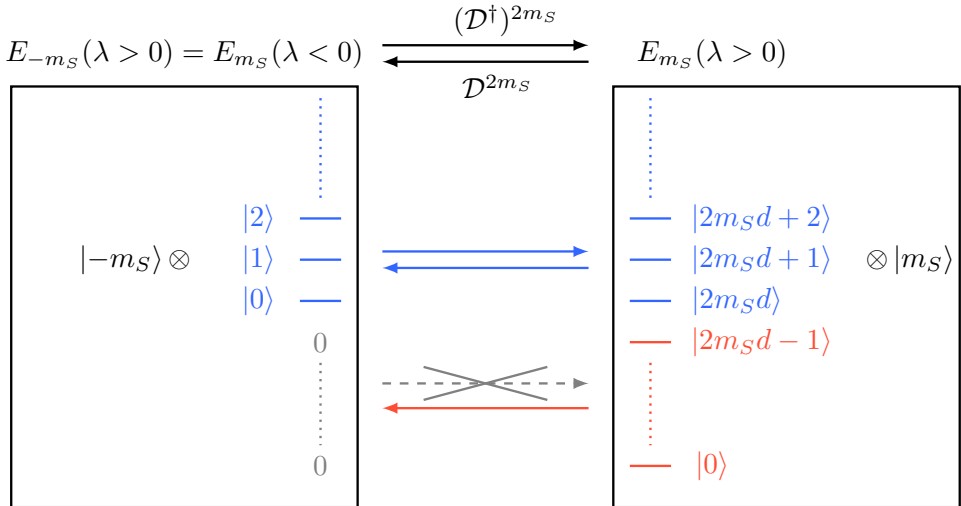

Figure 7: Sketch of the mappings $(\mathcal{D}^\dagger)^{2m_S}$ and $\mathcal{D}^{2m_S}$ between the two subspaces $E_{-m_S}$ and $E_{m_S}$. Reversing $\lambda$ amounts to reverse $m_S$. The branch $m_S$ at $\lambda > 0$ has $2m_S d$ modes that do not have a paring partner for $\lambda < 0$. There are therefore $2m_S d$ extra modes in the branch $m_S$ for $\lambda > 0$ compared to $\lambda < 0$ (in red). All the other number states can be paired with a number state of the space $E_{-m_S}$, or equivalently $E_{m_S}$ with $\lambda < 0$.

invertible. Here $\mathcal{D}^{2m_S}$ is Fredholm since it is invertible between $E_{m_S}$ and $E_{-m_S}$, up to a finite number ($2m_S d$) of elements, and we have

$$\Delta \mathcal{N}_{m_S} = \text{ind}\mathcal{D}^{2m_S}\,, \tag{118}$$

with

$$\text{ind}\mathcal{D}^{2m_S} = 2m_S d\,, \tag{119}$$

in the present case. The values of this Fredholm index are found to correspond to those of the Chern numbers $\mathcal{C}_{m_S}$ calculated in (104) for the corresponding symbol Hamiltonian, for arbitrary spins $S$ and integer $d$, which thus verifies explicitly the Atiyah-Singer theorem in that case, or equivalently the monopole - spectral flow correspondence for this class of $h.\hat{S}$ models.[7] The Berry-Chern monopole-spectral flow correspondence then naturally follows from (118) for this class of $h.\hat{S}$ models.

In the next section, we diagonalize $\hat{\mathcal{H}}_{\text{op}}$ to compute explicitly the spectral flows and their associated eigenstates.

## 3.4 Eigenstates of $\hat{\mathcal{H}}_{\text{op}}$ and verification of the monopole - spectral flow correspondence

In this section, we aim at diagonalizing the operator Hamiltonian $\hat{\mathcal{H}}_{\text{op}}$ as $\hat{\mathcal{H}}_{\text{op}}|\Psi\rangle = \tilde{E}|\Psi\rangle$ and plotting its eigenvalues spectrum $\tilde{E}$ as a function of $\lambda$ beyond the Dirac case $S > 1/2$ and $d = 1$, in order to check explicitly the monopole - spectral-flow correspondence as predicted from the values of the Chern numbers (104) and the analytical index (119). For that purpose,

---

[7]The $-$ sign difference between the analytical index and the Chern number comes from the convention $\mathcal{A} = +\text{i}\langle\psi|\,\text{d}\,|\psi\rangle$ of the Berry connection. They would have the same sign if one took instead the condensed matter convention $\mathcal{A} = -\text{i}\langle\psi|\,\text{d}\,|\psi\rangle$.

we propose the following anzatz for the eigenstates

$$\text{Anzatz } A0: \qquad |\Psi_0\rangle = \sum_{m_S=-S}^{S} \psi_{m_S} |(S+m_S)d+n, m_S\rangle, \qquad (120)$$

where again $|n, m_S\rangle \equiv |n\rangle \otimes |m_S\rangle$ where $|n\rangle$ is a number state, i.e. that satisfies $\hat{a}^\dagger \hat{a} |n\rangle = n |n\rangle$ and $|m_S\rangle$ is a spin state, i.e. it satisfies $\hat{S}_z |m_S\rangle = m_S |m_S\rangle$. For each value of $n$, the anzatz $A0$ is a superposition of such $2S+1$ base states. If we wrote the spin operators in a matrix form, the anzatz $A0$ would take the form of a vector of number states that would read

$$\begin{pmatrix} \psi_S & |(2S)d+n\rangle \\ \psi_{S-1} & |(2S-1)d+n\rangle \\ \vdots & \vdots \\ \psi_{-S+1} & |d+n\rangle \\ \psi_{-S} & |n\rangle \end{pmatrix}. \qquad (121)$$

Consistently, one recovers the solution of the 2D Dirac and 3D Weyl operator Hamiltonians (8) for the branches $\pm$ when $S = 1/2$ and $d = 1$.

By applying $|\Psi_0\rangle$ to $\hat{\mathcal{H}}_{\text{op}}$ and then projecting onto the states $\langle n+S-p, p|$, one obtained the eigenvalue equation

$$\underbrace{\begin{pmatrix} S\lambda & \gamma_S & & & & & \\ \gamma_S & & & & & & \\ & \ddots & \ddots & \ddots & & & \\ & & \gamma_{m_S+1} & m_S\lambda & \gamma_{m_S} & & \\ & & & \ddots & \ddots & \ddots & \\ & & & & & \gamma_{-S} & \\ & & & & \gamma_{-S} & -S\lambda \end{pmatrix}}_{H_n^{(2S+1)}} \begin{pmatrix} \psi_S \\ \psi_{S-1} \\ \vdots \\ \psi_{m_S} \\ \vdots \\ \psi_{-S+1} \\ \psi_{-S} \end{pmatrix} = \tilde{E}_n^{(m_S)} \begin{pmatrix} \psi_S \\ \psi_{S-1} \\ \vdots \\ \psi_{m_S} \\ \vdots \\ \psi_{-S+1} \\ \psi_{-S} \end{pmatrix}, \qquad (122)$$

where we have introduced the coefficients

$$\gamma_{m_S} = \sqrt{S(S+1) - m_S(m_S-1)} \prod_{j=0}^{d-1} \sqrt{(S+m)d+n-j}, \qquad (123)$$

which depend on $S, m_S, n$ and $d$. The upper index $m_S$ in $\tilde{E}_n^{(m_S)}$, which is the spin projection along $z$, can be seen as the branch index, which accordingly varies from $-S$ to $S$ by integer or half integer values. The lower index $n$ labels the discrete levels into each branch $m_S$. The matrix $H_n^{(2S+1)}$ to be diagonalized has a size $(2S+1) \times (2S+1)$ and is defined for a fixed $n$. Its characteristic polynomial $P_n^{(2S+1)} \equiv \det(H_n^{(2S+1)} - \tilde{E}_n^{(m_S)} \text{Id})$ is therefore of degree $2S+1$ in $\tilde{E}^{(m_S)}$, and has real coefficients.[8] For each $n \in \mathbb{N}$, the secular equation $P_n^{(2S+1)} = 0$ yields $2S+1$ real solutions, corresponding to a given mode $n$ in each of the energy branches. The entire spectrum of $\hat{\mathcal{H}}_{\text{op}}$ is finally the list $\{\tilde{E}_n^{(m_S)}\}$ for all $n \in \mathbb{N}$ and for each branch $m_S$. Similarly, the coefficients $\psi_{m_S}$ should also depend on $n$. To simplify the notation, and because it is not crucial in the computation of the eigenvalues spectra, we will abusively write $\psi_{m_S}$ to refer to amplitudes with possibly different $n$. The spectra obtained for different values of $S$ and $d$ with the anzatz $A0$ are shown in figure 8.

---

[8]Do not confuse the *degree* of the polynomial, which is simply the higher power of its variable, with the *degree* $\deg h$ of the Hamiltonian map.

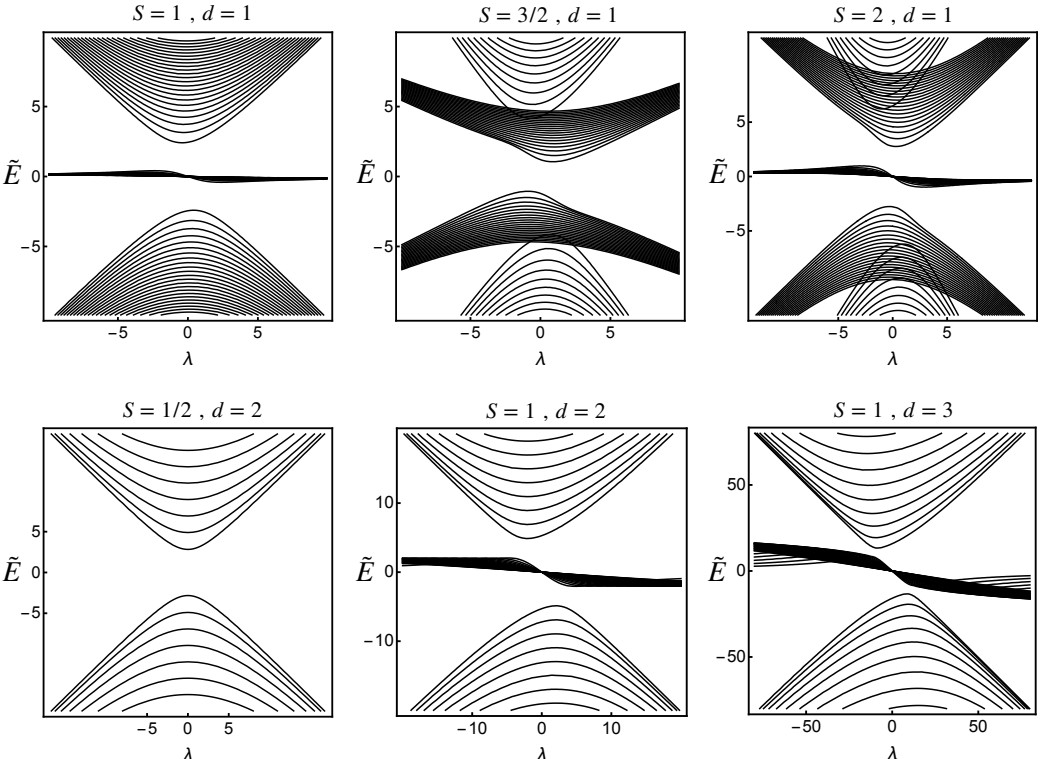

Figure 8: A few examples of eigenvalue spectra $\tilde{E}$ of $\hat{\mathcal{H}}_{\text{op}}$ obtained with the anzatz $A0$ in (120) beyond the case $S = 1/2$ and $d = 1$.

None of these spectra manifest a spectral flow. The solutions of $\hat{\mathcal{H}}_{\text{op}}$ that do develop a spectral flow follow from different anzatz than $A0$ and can be constructed separately one by one. To do so, we notice that the lowest mode is reached for $n = 0$, and has the following form when using the matrix representation for spins degrees of freedom

$$
\begin{pmatrix}
\psi_S & |(2S)d\rangle \\
\psi_{S-1} & |(2S-1)d\rangle \\
\vdots & \vdots \\
\psi_{-S+1} & |d\rangle \\
\psi_{-S} & |0\rangle
\end{pmatrix}.
\tag{124}
$$

The particularity of $|\Psi_0\rangle$ is that it decomposes as a superposition over the $2S + 1$ spin states $|m_S\rangle$. Actually, other eigenstates can be constructed, and decomposed over $2S + 1 - p$ spin states $|m_S\rangle$ with $p > 1$. The lower number state $|0\rangle$ being associated with the lower spin state $|-S\rangle$, a first iteration for $p = 1$ then consists in replacing this number state by $|0\rangle \rightarrow 0$ and reducing the other number states, associated to all the higher spin states $|m_S > -S\rangle$, to $n = -1$ as

$$
\begin{pmatrix}
\psi_S^{(1)} & |(2S)d-1\rangle \\
\psi_{S-1}^{(1)} & |(2S-1)d-1\rangle \\
\vdots & \vdots \\
\psi_{-S+1}^{(1)} & |d-1\rangle \\
0 &
\end{pmatrix}.
\tag{125}
$$

Such a form will be referred to as the anzatz $A1$ as it possesses 1 zero in its decomposition over spin states, and we can write

$$\text{Anzatz } A1: \qquad |\Psi_1\rangle = \sum_{m_S=-S+1}^{S} \psi_{m_S}^{(1)} |(S+m_S)d-1, m_S\rangle \,. \qquad (126)$$

This new Anzatz yields a new eigenvalues problem where the matrix $H_{-1}^{2S}$ to be diagonalized is obtained from $H_n^{(2S+1)}$ by removing the last line and the last column in (122) and by substituting $n=-1$. It yields $2S$ energy levels that connect the energy branches obtained previously with anzatz $A0$. The procedure must be iterated $d$ times by lowering $n$ until the lower number state $|0\rangle$ is reached as

$$\begin{pmatrix} \psi_S^{(1)} & |(2S)d-d\rangle \\ \psi_{S-1}^{(1)} & |(2Sd-2d)\rangle \\ \vdots & \vdots \\ \psi_{-S+1}^{(1)} & |0\rangle \\ 0 & \end{pmatrix}, \qquad (127)$$

giving at each iteration a different spectral flow between the same branches $m_S$ and $m_S'$. Once the spin component $m_S = -S+1$ is finally given by the number state $|n=0\rangle$, one iterates the anzatz to look for solutions where both the spin components $m_S = -S$ and $m_S = -S+1$ are zero. More generally, the anzatz with $p$ zeros in the decompositions over the $p$ lower spin states reads

$$\text{Anzatz } Ap: \qquad |\Psi_p\rangle = \sum_{m=-S+p}^{S} \psi_{m_S}^{(p)} |(S+m_S)d-p, m_S\rangle \,. \qquad (128)$$

One applies this algorithm by keeping decreasing the number states until all the spin components are vanishing but the $m_S = S$ one as

$$\begin{pmatrix} |d-1\rangle \\ 0 \\ \vdots \\ 0 \end{pmatrix}, \qquad (129)$$

that is

$$\text{Anzatz } A(2S-1): \qquad |\Psi_{2S-1}\rangle = |d-1, S\rangle \,. \qquad (130)$$

The number state $|d-1\rangle$ can still be decreased $d-1$ times to reach the final possibility with the fundamental $|0\rangle$ for the spin $m_S = S$ and 0 amplitude for every other spin components. Those $d-1$ 'sub-anzatz', so to speak, all yield the same eigenenergy spectrum with a non-dispersive branch $\tilde{E} = S\lambda$. This branch is common to all spectra for any values of $S$ and $d$, and is the only one existing in the Dirac case for $S = 1/2$ and $d = 1$.

The complete spectra, that now display various spectral flows, are shown in figure 9. Importantly, all those spectral flows satisfy the correspondence with the Berry-Chern monopole of the symbol Hamiltonian as $\mathcal{N}_{m_S} = -\mathcal{C}_{m_S} = 2m_S d$, where $\mathcal{N}_{m_S}$ is the number of modes that are gained by the band $m_S$ when sweeping $\lambda$ from $-\infty$ to $+\infty$.

After the canonical Dirac case ($S = 1/2$ and $d = 1$), the spin 1 case with $d = 1$ is the most encountered in the literature. It naturally appears in the linear rotating shallow water model describing rotating fluids such as oceans and atmosphere over large distances, with a varying Coriolis parameter along the latitude that changes sign at the equator [15, 81]. In that case, the Coriolis parameter plays the role of the mass term, and the spectral flow parameter $\lambda$ is

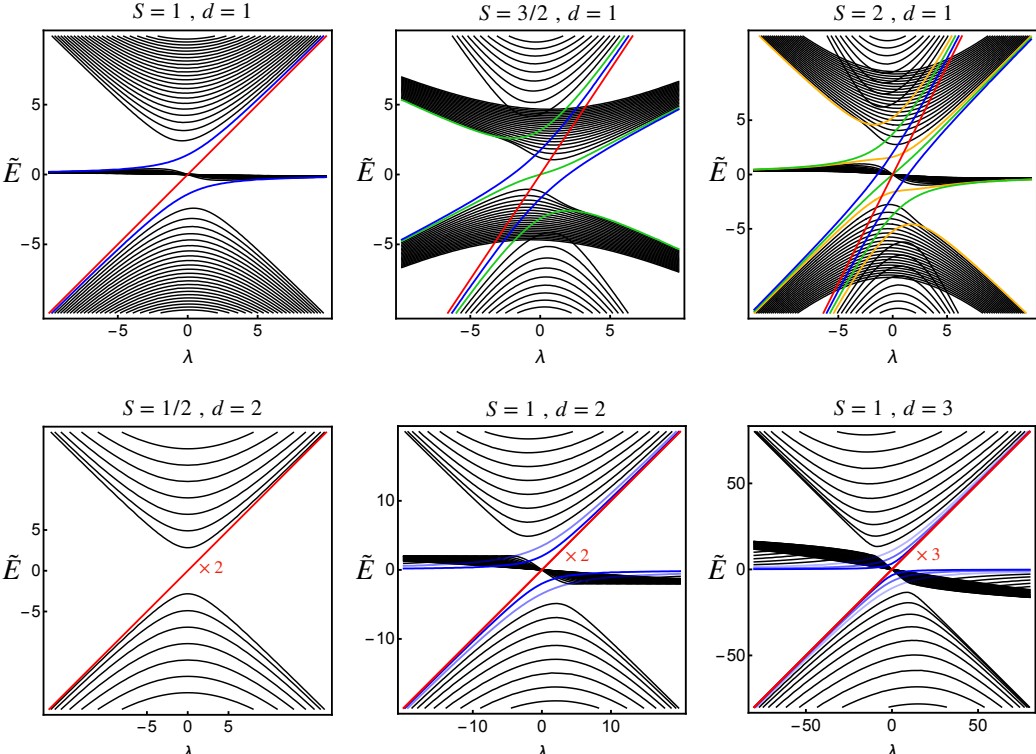

Figure 9: A few examples of full eigenvalue spectra $\tilde{E}$ of $\hat{\mathcal{H}}_{\mathrm{op}}$ beyond the case $S = 1/2$ and $d = 1$. The black branches are obtained with the anzatz 0 as in figure 8 and the colored spectral flows are obtained by applying the successive anzatz $Ap$ (128) : (red) $p = 2S$, (blue) $p = 2S - 1$, (green) $p = 2S - 2$ and (yellow) $p = 2S - 3$.

the wave vector parallel to the equator. The spectral flow then corresponds to unidirectional Eastward equatorial waves known as the Kelvin and the Yanai waves. The explicit mapping between $\hat{\mathcal{H}}_{\mathrm{op}}$ and the shallow water model is shown in Appendix C. A similar phenomenology appears with classical waves in various continuous systems such as electromagnetic waves in 2D gyrotropic media and plasmas, both in a varying magnetic field [61,82]. In those physically very different examples, the spectral flows are interpreted as interface sates along a line in real space where a varying mass term (the Coriolis force or the magnetic field) changes sign. As discussed at the beginning of this manuscript with the Weyl fermions, the same spectrum can emerge from a very different physical mechanism, that is the coupling of a spin 1 quantum particle in 3D with a magnetic field [27]. Other generalizations of Weyl fermions (i.e. beyond $S = 1/2$ and $d = 1$) have been discussed theoretically and experimentally in various materials [19,83,84].

# 4 Take Home message

Spectral flows are ubiquitous in wave physics and quantum mechanics. A simple method to look for spectral flows consists in searching for degeneracy points of the symbol Hamiltonian (i.e. of the dispersion relation), and then "quantize" the system by either varying in space a parameter controlling the gap amplitude (the mass term) such that it changes sign, or by applying a constant magnetic field if we one starts with massless charged quantum particles in 3D. The topological description of such spectral flows is based on a correspondence with a "semiclassical" description of the system, encoded through the symbol Hamiltonian, where

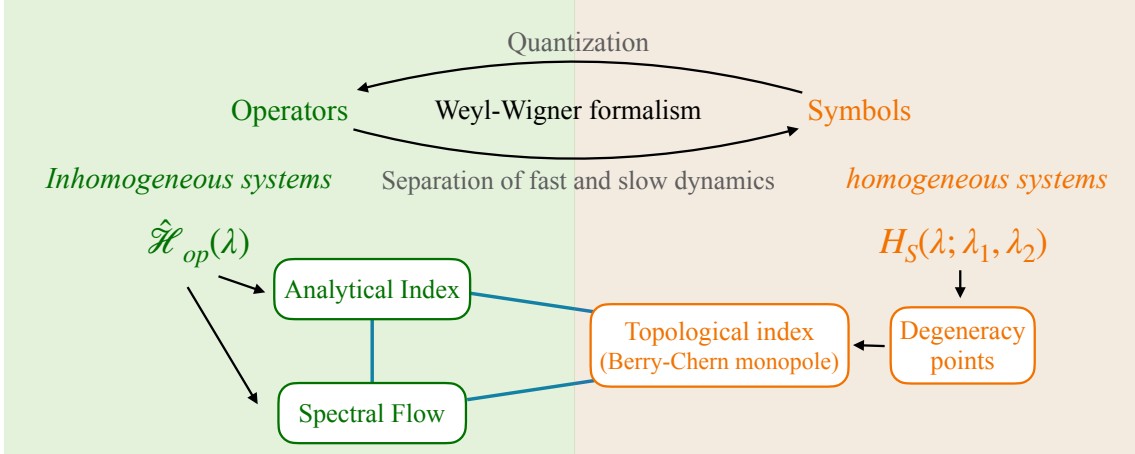

Figure 10: Quick overview of the general structure of the theory of the monopole-spectral flow correspondence. The blue lines represent an equality relation.

the mass term is assumed to vary sufficiently slowly, that is over a length scale much larger than the wave length of reference. This limit corresponds to a description where the system can be decoupled into two systems with a fast and a slow dynamics, the latest being treated "classically". The topological indices of the symbol Hamiltonians (the first Chern numbers), describing the system in that limit, appears as charges associated to the degeneracies of its eigenvalues in a three-dimensional parameter space, where two dimensions correspond to the classical phase space of canonical conjugate variables and the third dimension is given by the spectral flow parameter. For the class of models we have considered ($h.\hat{S}$ models), those Chern numbers are related to analytical indices of the operator Hamiltonian through the Atiyah-Singer theorem, which are themselves a measure of the spectral flow in each branch. The existence of a spectral flow does not automatically mean the existence of unidirectional modes, unless a direct gap separates different branches in the spectrum of the operator Hamiltonian. However, at least in the class of models discussed here, the modes participating to the spectral flow are 1) the most localized (in one direction) in space, and 2) those with the less nodes (the closest to the vacuum mode $|0\rangle$). They are in some sens localized at the "boundary" of the Fock space, if one interprets the states $|n\rangle$ as positions on a half infinite lattice labelled by $n$.

To summarize, the general structure of this theory is sketched in figure 10.

## Acknowledgements

I am thankful to Clément Tauber, Antoine Venaille and Frédéric Faure for inspiring discussions about various physical and mathematical aspects addressed in these lecture notes. This manuscrit was written partially from the lecture I gave at the École Normale Supérieure de Lyon from 2018, and in a much less detailed version at Les Houches and Cargèse summer schools in 2018 and 2019, and partially from my Habilitation thesis [74].

# A  Operator-Symbol correspondence

In quantum mechanics position and momentum are described by operators $\hat{x}$ and $\hat{p}$, while in Hamiltonian classical mechanics, those are scalar conjugate variables in phase space $(x, p)$. A crucial difference between the two theories being that $\hat{x}$ and $\hat{p}$ do not commute while $x$ and $p$ do. The mapping from phase space to Hilbert space of operators is called *quantization*. This procedure, which was at the root of the construction of quantum mechanics from classical mechanics during the first decades of the XXth century, stimulated many mathematical works. Indeed such a mapping is not unique, and many quantization procedures where developed called e.g. $x - p$ and $p - x$ quantizations, Wick and anti-Wick quantizations, geometric quantization, and so on. The inverse procedure, that is the mapping from Hilbert space to phase space of classical observables is also routinely used in physics, with for instance the WKB approximation, that consists in expanding a wave function with respect to a small parameter, and the Wigner transform that allows the representation of a quantum state in phase space. Those procedures are based on expansions with respect to a small parameter, namely the Planck constant $h$. Actually, such procedures are not restricted to quantum physics, but apply much more broadly to any field that deals with differential equations with a small parameter. This field is referred to as semi-classical analysis, or also micro-local analysis in the mathematical context. When an operator of Hilbert space is mapped to an observable on phase space, this resulting "classical" observable is called the *symbol*. So in the following, a symbol will refer to a function $\sigma$ on phase space, and will be associated to an operator $\mathrm{Op}[\sigma] \equiv \hat{\sigma}$ through a quantization procedure. Conversely, we shall use the notation $\sigma[\hat{\sigma}] = \sigma$ to indicate that we take the symbol of an operator.

In practice, we will be interested in cases where there is a one-to-one correspondence between a symbol and its operator. For this, we use a standard quantization procedure, called Weyl quantization. Weyl quantization turns a function $\sigma(x, p)$ on phase space into an operator $\hat{\sigma}$, through the expression of the action of this operator on an arbitrary wave function in position representation $\psi(x)$, called the Weyl transform and that reads

$$\hat{\sigma}\psi(x) = \frac{1}{2\pi\epsilon} \int_{\mathbb{R}} \mathrm{d}x' \int_{\mathbb{R}} \mathrm{d}p \; \sigma\left(\frac{x + x'}{2}, p\right) e^{ip(x-x')/\epsilon} \psi(x'). \tag{A.1}$$

This formula is given for 1D systems, but generalizes straightforwardly at any dimension. Note the existence of a parameter $\epsilon$ in the definition. Strictly speaking, a quantization procedure is only well-defined with respect to such a parameter, in order to recover a classical behavior of the dynamics in the limit $\epsilon \to 0$. The Weyl transform (A.1) yields

$$\sigma(x, p) = x \quad \to \quad \hat{\sigma}\psi(x) = x\psi(x), \tag{A.2}$$

$$\sigma(x, p) = p \quad \to \quad \hat{\sigma}\psi(x) = -i\epsilon\partial_x\psi(x), \tag{A.3}$$

which reproduces the expected quantization rule of position $x$ and momentum $p$, that is usually written as the substitution

$$x \quad \to \quad \mathrm{Op}[x] = \hat{x} = x, \tag{A.4}$$

$$p \quad \to \quad \mathrm{Op}[p] = \hat{p} = -i\epsilon\partial_x, \tag{A.5}$$

in position representation.

Conversely, the symbol $\sigma(x, p)$ of an operator $\hat{\sigma}(\hat{x}, \hat{p})$ can be obtained, at least formally, from the Wigner transform

$$\sigma(x, p) = \int \mathrm{d}x' \Sigma\left(\frac{x + x'}{2}, x - \frac{x - x'}{2}\right) e^{-ipx'/\epsilon}, \tag{A.6}$$

where $\Sigma(x, x')$ is the kernel of the operator $\hat{\sigma}$, that is, it verifies

$$\hat{\sigma}\psi(x) = \int \mathrm{d}x' \Sigma(x, x')\psi(x'). \tag{A.7}$$

Equations (A.1), (A.6) and (A.7) yield a self-consistent definition of the Wigner-Weyl calculus. Computing the symbol of an operator via the Wigner transform necessitates the knowledge of the Kernel of this operator, which, in general, is a distribution. This is not always the easiest way to compute the symbol. In practice, one can use the powerful formula of the Moyal product $\star$ between symbols $\sigma_1(x, p)$ and $\sigma_2(x, p)$

$$\sigma_1(x, p) \star \sigma_2(x, p) \equiv \sum_{n=0}^{\infty} \sum_{m=0}^{n} \frac{1}{n!} \left(\frac{\mathrm{i}\epsilon}{2}\right)^n (-1)^m \binom{n}{m} \left(\partial_x^{n-m}\partial_p^m \sigma_1\right)\left(\partial_p^{n-m}\partial_x^m \sigma_2\right), \tag{A.8}$$

that is related to the product of their operators as

$$\hat{\sigma}_1\hat{\sigma}_2 = \mathrm{Op}[\sigma_1 \star \sigma_2]. \tag{A.9}$$

Using the fact that identity verifies $\mathrm{Op}[\mathbb{I}] = \mathbb{I}$, the symbol of the operators $\hat{x}$ and $\hat{p}$ are then easily obtained by using (A.8) and (A.9) and one gets

$$x \quad \leftarrow \quad \mathrm{Op}[x] = \hat{x} = x, \tag{A.10}$$
$$p \quad \leftarrow \quad \mathrm{Op}[p] = \hat{p} = -\mathrm{i}\epsilon\partial_x, \tag{A.11}$$

as expected. When $\epsilon \ll 1$, this formula can be used to approximate a product of operators by the operator of the expansion, in powers of $\epsilon$, of the product of their symbols. In particular, it yields that the first order correction in $\epsilon$ to $\hat{\sigma}_1\hat{\sigma}_2 = \mathrm{Op}[\sigma_1\sigma_2]$ is given by the operator of the Poisson bracket $\{\sigma_1, \sigma_2\}$, that is precisely the commutator $[\hat{\sigma}_1, \hat{\sigma}_2]$. For our purpose however, the operators we have at hand are polynomials in $\hat{x}$ and $\hat{p}$, so that the sum in the $\star$ product (A.8) yields an exact result with a finite number of terms. We shall consider all those terms, and thus take $\epsilon = 1$ from now. This yields the correspondence

$$x + \mathrm{i}p \quad \leftrightarrow \quad \hat{a}, \tag{A.12}$$
$$x - \mathrm{i}p \quad \leftrightarrow \quad \hat{a}^\dagger, \tag{A.13}$$

where $x$ and $p$ designate two canonical conjugate observables in phase space, not necessarily position and momentum. Note that the relation (A.9) is particularly useful to compute the symbol of a product of operators. By taking formally the symbol of each members of this equation, one has

$$\sigma[\hat{\sigma}_1\hat{\sigma}_2] = \sigma_1 \star \sigma_2. \tag{A.14}$$

We can use the $\star$ product to establish the correspondences (86) and (87) between $H_S$ and $\hat{\mathcal{H}}_{\mathrm{op}}$ for the $h.\hat{S}$ model. To demonstrate this relation, let us proceed by induction. First, we know that

$$\mathrm{Op}[\alpha x + \beta p] = \alpha\hat{x} + \beta\hat{p}, \tag{A.15}$$

where $\alpha$ and $\beta$ are arbitrary complex numbers. Then we assume the relation

$$\mathrm{Op}[(\alpha x + \beta p)^{m-1}] = (\alpha\hat{x} + \beta\hat{p})^{m-1}, \tag{A.16}$$

for a given $m-1$. Let us now prove that this relation remains true when $m-1 \to m$. We have

$$(\alpha\hat{x} + \beta\hat{p})^m = (\alpha\hat{x} + \beta\hat{p})^{m-1}(\alpha\hat{x} + \beta\hat{p}) \tag{A.17}$$

$$= \mathrm{Op}[(\alpha x + \beta p)^{m-1}]\mathrm{Op}[\alpha x + \beta p] \tag{A.18}$$

$$= \mathrm{Op}[(\alpha x + \beta p)^{m-1} \star (\alpha x + \beta p)] \tag{A.19}$$

$$= \mathrm{Op}\Big[ \sum_{m,n=0}^{\infty} \frac{1}{n!}\left(\frac{\mathrm{i}\epsilon}{2}\right)^n (-1)^m \binom{n}{m}$$
$$\times \left(\partial_x^{n-m}\partial_p^m (\alpha x + \beta p)^{m-1}\right)\left(\partial_p^{n-m}\partial_x^m (\alpha x + \beta p)\right)\Big]. \tag{A.20}$$

The last term in (A.20) is nonzero only when $(n, m)$ takes the values $(0, 0)$, $(1, 0)$ and $(1, 1)$. The contribution $(0, 0)$ yields the term $(\alpha x + \beta p)$, while the two other contributions $(1, 0)$ and $(1, 1)$ give two terms proportional to $\epsilon$, such that

$$(\alpha\hat{x} + \beta\hat{p})^m = \mathrm{Op}[(\alpha x + \beta p)^{m-1}(\alpha x + \beta p)$$
$$+ \frac{\mathrm{i}\epsilon}{2}(m-1)\alpha(\alpha x + \beta p)^{m-2}\beta - \alpha(m-1)\beta(\alpha x + \beta p)^{m-2}]. \tag{A.21}$$

The two terms proportional to $\epsilon$ compensate each other, and we end up with

$$\mathrm{Op}[(\alpha x + \beta p)^m] = (\alpha\hat{x} + \beta\hat{p})^m, \tag{A.22}$$

which completes the proof. This formula is a famous result of Weyl calculus, that was first obtained long ago by McCoy before the $\star$ product was introduced [76]. Conversely, we also get that

$$\sigma[(\alpha\hat{x} + \beta\hat{p})^m] = (\alpha x + \beta p)^m, \tag{A.23}$$

which leads to the relations (86) and (87) when $\alpha = 1$ and $\beta = \pm\mathrm{i}$.

# B Toolbox of differential calculus: Stokes theorem and Brouwer degree formula

There are many good textbooks that give detailed and consistent introductions to differential forms, such as [26, 43, 85] that are dedicated to physicists. Here we present a digest summary that sketches a few basic definitions and results that are useful for the following.

Differential forms constitute a generalization of functions. A usual function $\lambda : \Lambda \to f(\lambda) \equiv \omega_0$ is then a 0-form, while its differential (provided it is differentiable) $\mathrm{d}f$ is an example of a 1-form. More generally, a 1-form is a kind of vector, called covector as it transforms as a covariant vector [85], and decomposes as $\omega_1 = \omega_1^i \mathrm{d}\lambda_i$ (where we use the implicit sum convention). If the dimension of parameter space $\Lambda$, that is our base space, is $N$, then the set of 1-forms $\Omega^1(\Lambda)$ is also of dimension $N$. A 2-form $\omega_2$ is an anti-symmetric tensor that decomposes as $\omega_2 = \omega_{jk}\mathrm{d}\lambda_j \wedge \mathrm{d}\lambda_k$ (with $j < k$), where the wedge product $\wedge$ generalizes the vector product defined in $\mathbb{R}^3$ to any dimension, by satisfying

$$\mathrm{d}\lambda_j \wedge \mathrm{d}\lambda_k = -\mathrm{d}\lambda_k \wedge \mathrm{d}\lambda_j, \tag{B.1}$$

and in particular $\mathrm{d}\lambda_j \wedge \mathrm{d}\lambda_j = 0$. By extension an r-form is a totally anti-symmetric tensor that decomposes as $\omega_r = \omega_{j_1, j_2, \cdots, j_r}^r \mathrm{d}\lambda_{j_1} \wedge \mathrm{d}\lambda_{j_2} \wedge \cdots \wedge \mathrm{d}\lambda_{j_r}$, and in particular the set $\Omega^N(\Lambda)$ of N-forms contains a single element $\omega_N = \omega_{1,2,\cdots,N}^N \mathrm{d}\lambda_1 \wedge \mathrm{d}\lambda_2 \wedge \cdots \wedge \mathrm{d}\lambda_N$.

As recalled at the beginning of this section, if $f$ is a 0-form (a function) then $\mathrm{d}f$ is a 1-form. This familiar result generalizes to any differential forms, with the *exterior derivative* on forms d that constitutes a map $\Omega^r(\Lambda) \xrightarrow{\mathrm{d}} \Omega^{r+1}(\Lambda)$; namely, the (exterior) derivative of an r-form is an (r+1)-form. In practice it is obtained as

$$\mathrm{d}\omega_r \equiv \frac{\partial}{\partial \lambda_{r+1}} \left( \omega^r_{j_1, j_2, \cdots, j_r} \right) \mathrm{d}\lambda_{r+1} \wedge \mathrm{d}\lambda_{j_1} \wedge \mathrm{d}\lambda_{j_2} \wedge \cdots \wedge \mathrm{d}\lambda_{j_r} \,. \tag{B.2}$$

Such an (r+1)-form, $\omega^{r+1} = \mathrm{d}\omega^r$ is called an *exact* form, as it is derived from an r-form (such as $\mathrm{d}f$). It may also happen that a form $\omega$ satisfies $\mathrm{d}\omega = 0$. This is called a *closed* form. Importantly, the anti-symmetric relation (B.1) imposes that $\mathrm{d}(\mathrm{d}\omega_r) = 0$, implying that any exact form is closed. The reciprocal is not true.

More generally, differential forms, such as the 2-form Berry curvature, are objects that one can integrate over manifolds. In particular, if the dimension of a manifold $M$ is $r$, then $\int_M \omega_r$ is a number. Of particular importance is the surface form $\Omega_{S^n}$, that is an n-form whose integral over a sphere $S^n$ embedded in $\mathbb{R}^{n+1}$ is 1. In cartesian coordinates, it reads

$$\Omega_{S^n} \equiv \sum_{i=1}^{n+1} (-1)^{n+1} \frac{x_i \mathrm{d}x_1 \wedge \cdots \wedge \mathrm{d}x_{i-1} \wedge \mathrm{d}x_{i+1} \wedge \cdots \mathrm{d}x_{n+1}}{\gamma_n \left( x_1^2 + x_2^2 \cdots + x_{n+1}^2 \right)^{(n+1)/2}} \,, \tag{B.3}$$

where $\gamma_n = 2\pi^{\frac{n+1}{2}} / \Gamma(\frac{n+1}{2})$ with $\Gamma(x)$ the Euler function, is the surface of the unit sphere $S^n$. Of particular importance for the following are the two examples in $\mathbb{R}^2$ and $\mathbb{R}^3$

$$\Omega_{S^1} = \frac{x\mathrm{d}y - y\mathrm{d}x}{2\pi (x^2 + y^2)} \quad \text{and} \quad \Omega_{S^2} = \frac{x\mathrm{d}y \wedge \mathrm{d}z + y\mathrm{d}z \wedge \mathrm{d}y + z\mathrm{d}x \wedge \mathrm{d}y}{4\pi (x^2 + y^2 + z^2)^{3/2}} \,. \tag{B.4}$$

A key result in differential calculus is Stokes theorem, that relates the integral of a differential form $\omega$ with that of its derivative $\mathrm{d}\omega$, *when it exists everywhere over $M$*, as

$$\text{Stokes theorem} \quad : \qquad \int_M \mathrm{d}\omega = \int_{\partial M} \omega \,, \tag{B.5}$$

where $\partial M$ denotes the boundary of the manifold $M$.

The expression (40) of the Chern number is very general. In particular, it does not depend on the model at hand. In order to derive an explicit expression for the Berry-Chern monopole model, it is useful to introduce the Brouwer theorem, that relates the integrals of a differential form over two different manifolds $A$ and $B$ of same dimension $n$. Consider a differential form $\omega$ defined over the manifold $B$ of local coordinates $\mathbf{y} = (y_1, \cdots, y_n)$, i.e. $\omega = \omega_{j_1, \cdots, j_n}(\mathbf{y})\mathrm{d}y_{j_1} \wedge \cdots \wedge \mathrm{d}y_{j_n}$, and a map $f$ between the two manifolds $\mathbf{x} : A \to \mathbf{y} = f(\mathbf{x}) \in B$. Then one can define a form on $A$ from $\omega$, through the action of $f$. It is called the *pull-back* of $\omega$ and reads

$$f^\star \omega = \omega_{j_1, \cdots, j_n}(f(\mathbf{y})) \det\left( \frac{\partial y^\alpha}{\partial x^\beta} \right) \mathrm{d}x_{j_1} \wedge \cdots \wedge \mathrm{d}x_{j_n} \,, \tag{B.6}$$

where the Jacobian matrix accounts for the change of local coordinates.

Then one has the following

$$\text{Brouwer theorem:} \qquad \int_A f^\star \omega = \deg f \int_B \omega \,, \tag{B.7}$$

where the degree of $f$ was defined in (79). An important application of the Brouwer theorem is that it gives an integral formulation of the degree of a map, which one obtained when applying the formula (B.7) when $B = S^n$ for the surface form $\Omega_{S^n}$.

For instance, in the case of the punctured plane $B = \mathbb{R}^2 \backslash \{0\} \cong S^1$, one gets

$$\deg h = \int_{\lambda \in S^1} h^\star \Omega_{S^1} \tag{B.8}$$

$$= \int_{\lambda \in S^1} \frac{1}{2\pi h^2} \left( h_x \mathrm{d}h_y - h_y \mathrm{d}h_x \right) . \tag{B.9}$$

Interpreting $\mathbf{h} = (h_x, h_x) = (x(t), y(t))$ as a vector position that depends on a parameter $t \in S^1$, one finds the standard expression of the winding number.

The degree is of main interest to characterize the topology of Berry monopole Hamiltonians i.e. that read $H = \boldsymbol{h}(\boldsymbol{\lambda}).\hat{\mathbf{S}}$, since the vector $\boldsymbol{h}$ defines a map $S^2 \overset{h/|h|}{\to} S^2$ whose degree is an integer that reads

$$\deg h = \int_{\lambda \in S^2} h^\star \Omega_{S^2} \tag{B.10}$$

$$= \int_{\lambda \in S^2} \frac{1}{4\pi h^3} \left( h_x \mathrm{d}h_y \wedge \mathrm{d}h_z + h_y \mathrm{d}h_z \wedge \mathrm{d}h_x + h_z \mathrm{d}h_y \wedge \mathrm{d}h_x \right) , \tag{B.11}$$

where the $h_i$'s are functions of $\lambda_\alpha$ so that $\mathrm{d}h_i = \frac{\partial h_i}{\partial \lambda_\alpha} \mathrm{d}\lambda_\alpha$, and one gets

$$\deg h = \frac{1}{4\pi} \int_{\lambda \in S^2} \frac{\epsilon^{ijk}}{h^3} h_i \frac{\partial h_j}{\partial \lambda_\alpha} \frac{\partial h_k}{\partial \lambda_\beta} \, \mathrm{d}\lambda_\alpha \wedge \mathrm{d}\lambda_\beta . \tag{B.12}$$

Finally, using the usual vectorial notation in $\mathbb{R}^3$, this expression takes the form

$$\deg h = \frac{1}{4\pi} \sum_{\alpha,\beta} \frac{1}{2} \int_{S^2} \frac{\mathbf{h}}{h^3} \cdot \left( \frac{\partial \mathbf{h}}{\partial \lambda_\alpha} \times \frac{\partial \mathbf{h}}{\partial \lambda_\beta} \right) \mathrm{d}\lambda_\alpha \wedge \mathrm{d}\lambda_\beta \tag{B.13}$$

This formula has a direct meaningful geometrical interpretation : it tells that the degree of the maps $S^2 \overset{h}{\to} S^2$ is a wrapping number, meaning that it counts the number of times the vector $\mathbf{h}/h = \mathbf{n}$ wraps the target sphere $S^2$ when $\boldsymbol{\lambda}$ spans the entire base space $S^2$.

## C The equatorial shallow water model and its analytical indices from the $h.\hat{S}$ model

Here we come back on the celebrated shallow water model that describes equatorial fluids waves in the Earth's atmosphere and oceans. Close to the equator, the modes of frequency $\tilde{\omega}$ and azimutal wave number $k_x$ (along the equator), around a state of rest, are given by the following equation [81, 86]

$$\tilde{\omega}\Psi = \mathcal{H}_{SW}\Psi , \tag{C.1}$$

with

$$\mathcal{H}_{SW} = \begin{pmatrix} 0 & -\mathrm{i}f(y) & ck_x \\ \mathrm{i}f(y) & 0 & c\mathrm{i}\partial_y \\ ck_x & c\mathrm{i}\partial_y & 0, \end{pmatrix} \tag{C.2}$$

in the basis of the perturbed velocity fields in $x$ and $y$ direction and the perturbed height elevation $(\delta u, \delta v, \delta h)$, where $c = \sqrt{gH}$ is the celerity of gravity waves with $g$ the standard gravity

and $H$ the height of the fluid at rest, and $f(y)$ is called the Coriolis parameter: it accounts for the Coriolis force that depends on the latitude. It has the dimension of a frequency (it is proportional to Earth angular rotation $\Omega$) and changes sign at the equator. It is convenient to work in the tangent plane picture, where the latitude dependence of the Coriolis parameter reads in terms of the $y$ coordinate that points toward the North pole in a tangent plane to Earth. In the vicinity of the equator, the Coriolis parameter can be approximated by $f(y) \sim 2\Omega y/R \equiv \beta y$ where $R$ is the Earth radius. This simplification is called the $\beta$-plane approximation. The operator Hamiltonian $\mathcal{H}_{SW}$ is known to display a spectral flow $\Delta\mathcal{N}_\pm = \pm 2$ and $\Delta\mathcal{N}_0 = 0$ for the three branches $(-1, 0, 1)$, and the Chern numbers of its symbol are accordingly $\mathcal{C}_\pm = \mp 2$ and $\mathcal{C}_0 = 0$ [15]. Here we make explicit its form as a particular case of the $h.\hat{S}$ model for $d = 1$ and $S = 1$ and thus gets its analytical indices.

For that purpose, let us notice that $\mathcal{H}_{SW}$ decomposes as

$$\mathcal{H}_{SW} = ci\partial_y \hat{S}_1 + f(y)\hat{S}_2 + ck_x \hat{S}_3, \tag{C.3}$$

where the $SU(2)$ commutation relations $[\hat{S}_1, \hat{S}_2] = i\hat{S}_3$, $[\hat{S}_2, \hat{S}_3] = i\hat{S}_1$ and $[\hat{S}_3, \hat{S}_1] = i\hat{S}_2$. From this point, it is clear that the shallow water model falls in the category of $h.\hat{S}$ models with $S = 1$ and $d = 1$, and thus its symbol Hamiltonian has the Chern numbers $\mathcal{C}_m = -2m$ with $m = -1, 0, 1$. To construct the index of $\mathcal{H}_{SW}$, one needs to transform it into $\hat{\mathcal{H}}_{\text{op}}$, which can be done by a unitary transformation

$$U\mathcal{H}_{SW}U^\dagger = ci\partial_y \hat{S}_x + f(y)\hat{S}_y + ck_x \hat{S}_z, \tag{C.4}$$

where $\hat{S}_z$ is diagonal. We find such a unitary operator to be

$$U = \frac{1}{\sqrt{2}}\begin{pmatrix} 1 & 0 & 1 \\ 0 & \sqrt{2} & 0 \\ -1 & 0 & 1 \end{pmatrix}, \tag{C.5}$$

and one gets

$$U\mathcal{H}_{SW}U^\dagger = \begin{pmatrix} k_x c & -i\frac{1}{\sqrt{2}}\frac{2\Omega}{R}y - c\partial_y & 0 \\ i\frac{1}{\sqrt{2}}\frac{2\Omega}{R}y - c\partial_y & 0 & -i\frac{1}{\sqrt{2}}\frac{2\Omega}{R}y - c\partial_y \\ 0 & i\frac{1}{\sqrt{2}}\frac{2\Omega}{R}y - c\partial_y & -k_x c \end{pmatrix}. \tag{C.6}$$

This rotated shallow water Hamiltonian has the same spectrum as the original one, owing to the unitarity of $U$.

There are two natural length scales in this problem : the Earth radius $R$ and the Rossby deformation radius $L \equiv \frac{c}{2\Omega}$ which gives the scale above which rotation is relevant. From this two lengths, one can construct the parameter $\alpha \equiv L/R$, which is small in our approach, and the characteristic length $\ell \equiv \sqrt{RL}$ that allows us to adimensionalize the shallow water Hamiltonian by introducing $\hat{a} = \frac{1}{\sqrt{2}}(\frac{y}{\ell} + \ell\partial_y)$ and $\hat{a}^\dagger = \frac{1}{\sqrt{2}}(\frac{y}{\ell} - \ell\partial_y)$ so that

$$U\mathcal{H}_{SW}U^\dagger = 2\Omega\begin{pmatrix} \lambda & -i\alpha\hat{a}^\dagger & 0 \\ i\alpha\hat{a} & 0 & -i\alpha\hat{a}^\dagger \\ 0 & i\alpha\hat{a} & -\lambda \end{pmatrix}, \tag{C.7}$$

with $\lambda = \frac{k_x c}{2\Omega}$, which also reads

$$\mathcal{H}_{SW} = 2\Omega U^\dagger\left(-i\alpha\hat{a}^\dagger\hat{S}_+ + i\alpha\hat{a}\hat{S}_- + \lambda\hat{S}_z\right)U. \tag{C.8}$$

From here, we can introduce the indices $\text{ind}(\mathcal{D}_{SW}^{2m_S})$ of the shallow water model with $\mathcal{D}_{SW} = i\alpha U^\dagger \hat{a}\hat{S}_- U$ Since the index is invariant by homotopy, one gets

$$\text{ind}(\mathcal{D}_{SW}^{2m_S}) = \text{ind}(U^\dagger \hat{a}\hat{S}_- U)^{2m_S} \tag{C.9}$$

and since it is also invariant by a unitary transformation, one finally finds

$$\text{ind}(\mathcal{D}_{SW}^{2m_S}) = \text{ind}(\mathcal{D}^{2m_S}) = 2m_S, \tag{C.10}$$

with $m_S = 0, \pm 1$. This is precisely the index of the $h.\hat{S}$ model for $d = 1$ and $S = 1$, which yields the spectral flows

$$\Delta\mathcal{N}_\pm = \pm 2 \quad \text{and} \quad \Delta\mathcal{N}_0 = 0, \tag{C.11}$$

as expected.

We now express the corresponding eigenmodes of the spectral flow. Following the procedure detailed in the section 3.4, one can construct the two eigenstates for the spectral flow in the case $S = 1$ and $d = 1$: $|\Psi_1\rangle$ with 1 zero component and $|\Psi_2\rangle$ with 2 zero components in the $|m_S\rangle$ basis, as

$$|\Psi_2\rangle = \begin{pmatrix} |1\rangle \\ 0 \\ 0 \end{pmatrix}, \qquad |\Psi_1\rangle = \begin{pmatrix} \psi_1|1\rangle \\ |0\rangle \\ 0 \end{pmatrix}, \tag{C.12}$$

where $\psi_1$ is a coefficient. The profiles in the $y$ direction of the eigenstates for the spectral flow of the original shallow water model in the $\beta$-plane have therefore the following form

$$\langle y|U^\dagger \Psi_2\rangle = \begin{pmatrix} 1 \\ 0 \\ -1 \end{pmatrix} e^{-\frac{1}{2}(\frac{y}{\ell})^2} \equiv \Psi_K(y) \tag{C.13}$$

$$\langle y|U^\dagger \Psi_1\rangle = \begin{pmatrix} y \\ \psi_1 \\ -y \end{pmatrix} e^{-\frac{1}{2}(\frac{y}{\ell})^2} \equiv \Psi_Y(y). \tag{C.14}$$

The mode $\Psi_K$ corresponds to a longitudinal velocity field $\delta u$ and a height elevation $\delta h$ which are symmetric with respect to the latitude $y$, while the velocity perpendicular to the equator $\delta v$ is zero. It corresponds to the well-known equatorial Kelvin mode. The second mode, $\Psi_Y$, has a longitudinal velocity field $\delta u$ and a height elevation $\delta h$ which are antisymmetric with respect to the latitude $y$, and the velocity perpendicular to the equator $\delta v$ is nonzero and symmetric. It is known as the Yanai or mixed planetary-gravity mode [81, 86].

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
