# Peer review of "Berry-Chern monopoles and spectral flows"

_SciPost Physics Lecture Notes, doi:SciPost Phys. Lect. Notes 39 (2022)_

## Round 1 · Referee Report · Bruno Mera · 2021-12-21

Strengths

1-Relevance of the subject to the field of topological phases and wave physics;
2-Pedagogical character of the text;
3-Concrete, very interesting applications of the "spectral flow-monopole correspondence", ranging from well-known examples from condensed matter, such as 2D Dirac fermions, 3D Weyl semi-metals, to lesser known examples like the equatorial shallow water model---showing how topological physics appears in many systems.

Weaknesses

None

Report

The present lecture notes provide an extensive pedagogical exploration of what the author refers to as the "Berry-Chern monopole correspondence", which is a relation between spectral flows and Chern numbers associated to degeneracy points. The concept is introduced through the example of two-band systems, but the author goes beyond this case, through the $h\cdot\hat{S}$ model , where the monopole-spectral flow correspondence is a manifestation of the Atiyah-Singer index theorem for a particular Fredholm operator. The relation between spectral flow and the symbol of the Hamiltonian---the latter being an adiabatic model of the former, where the "slow variables" are replaced by classical variables--- is thoroughly explored and rendered intelligible for non-experts.

The lecture notes are, in my opinion, very timely, very well-written and of ongoing interest to the physics community in a multidisciplinary way. The presentation of the material is very nice, with several concrete examples and figures illustrating the abstract concepts. It was quite pleasant to read the lecture notes and I am happy to recommend the lectures to publication in SciPost Physics Lecture Notes.

Below I present a list of typos and suggested changes to be considered by the author.

Requested changes

1. Page 4 "masse"->"mass"

2. Page 7 "separatelly" ->"separately"

3. Page 10. In the sentence:
"Such maps, between two closed orientable manifolds, (here two suraces) are classified by an integer-valued number, called degree"
I would write closed oriented (otherwise the sign of the degree is not well-defined) manifolds of the same dimension (which you later mention, but it's good to have it here since it's an important ingredient for the notion of degree).

4. In page 12 (and continue using it later on), you write that a fiber bundle is said to be "topological" when it is not isomorphic to a trivial bundle. I believe the common terminology in mathematics is to say that the bundle is "topologically non-trivial". Although I understand in physics usually the term "topological" refers to "topologically non-trivial".

5. In page 14, where it reads "$\mathcal{F}\to\mathcal{F}+d\mathcal{F}=\mathcal{F}$", I believe it should read "$\mathcal{F}\to\mathcal{F}-d^2\chi=\mathcal{F}$".

6. Page 15, there's an extra "are" in the sentence"For that reason, degeneracy points (or band crossing points) are play".

7. In page 21, the part of the sentence "and the fiber bundle defined over this base space cannot always be compactified (into a torus or a sphere for instance)" can lead to incorrect interpretations. I understand that you want a Hamiltonian that satisfies certain boundary conditions (for example periodic boundary conditions or certain conditions at infinity) in order to ensure that you have a well-defined fiber bundle over a compact space. But what is really being compactified is the base space and not the fiber bundle itself (you write "the fiber bundle cannot always be compactified into a torus or a sphere"). The question is whether the bundle that you had over the initial non-compact space gives rise or not to the compactification.

8. Page 21, "That is, One..." -> "That is, one".

9. Page 26, Eq. (78), the $\in$ symbols should be $\subset$ symbols. Also, there is a citation below the equation that is not compiling correctly.

10. Page 28, below Eq. (88c), "compare" -> "compared".

11. Page 36, "ligne"->"line".

12. Page 37, "fondamental"-> "fundamental".

13. Page 51, $\alpha\equiv=$->"$\alpha\equiv$".

  • validity: high
  • significance: high
  • originality: high
  • clarity: high
  • formatting: excellent
  • grammar: excellent

Author:  Pierre Delplace  on 2022-01-07  [id 2074]

(in reply to Report 1 by Bruno Mera on 2021-12-21)
Category:
remark

I thank very much the referee for its enthusiasm about this manuscript and for its careful reading of it.
I also thank him for his comments and agree with all of them, up to a minor detail about point 9. It is true that the vector h lies ''inside'' the surface in the figure 6, but this is not always the case, especially when the degree is 0. What I meant is that a point $\lambda$ is sent to $h(\lambda)$, so $h(\lambda)$ must be understood as a point that belongs to the new surface, by construction. I have rephrazed this part to avoid any confusion, I hope.

Bruno Mera  on 2022-01-10  [id 2083]

(in reply to Pierre Delplace on 2022-01-07 [id 2074])
Category:
remark

Indeed, I agree and understand what the author means. Perhaps it wasn't clear what I meant also -- it was simpler than that. The correction I intended was $\lambda \in \mathcal{S}_{\lambda_0}\subset \mathbb{R}^3$ and $h(\lambda)\in \mathcal{S}_{h}\subset\mathbb{R}^3$ (the second symbols in the version I have read before were $\in$ symbols instead of $\subset$ symbols, i.e. $ \lambda \in \mathcal{S}_{\lambda_0}\in \mathbb{R}^3$ and $h(\lambda)\in \mathcal{S}_{h}\in\mathbb{R}^3$), as in the surfaces $\mathcal{S}_{\lambda_0}$ and $\mathcal{S}_{h}$ are both subsets of $\mathbb{R}^3$. I apologize for not being more specific before and I hope it is clear now.

Anonymous on 2022-01-11  [id 2086]

(in reply to Bruno Mera on 2022-01-10 [id 2083])

Oh yes, you are right. I have now made the change of notation accordingly. Thank you for noticing this point.

Anonymous on 2022-01-11  [id 2088]

(in reply to Anonymous Comment on 2022-01-11 [id 2086])

You are very welcome.

---

## Round 1 · Referee Report · Anonymous · 2021-12-27

Strengths

1. Accessible for nonspecialists
2. Require minimum mathematical background
3. Useful for readers in various fields

Weaknesses

None

Report

This lecture note provides a clear description of “the spectral flow” in several physically interesting situations: two-dimensional massive Dirac fermions with the sign change, 3D Weyl semimetal subject to a magnetic field, “Haldane model” with an interface, and 2D polaritons with an interface. They are described in a unified way using a matrix Hamiltonian of a bosonic operator. It can be accessible for non-specialists. Then based on the “operator-symbol correspondence”, the topology of the “symbol” Hamiltonian is discussed by using Berry connection. This part is also well written in a compact way.
The scheme is generalized to the spin-operator coupled model and its relation to the index theorem is discussed by calculating the topological index explicitly. In appendix C, the spectral flow of the shallow water model, describing equatorial fluids waves in the Earth, is discussed as an example.
The manuscript is clearly written and provides key ideas of topological physics for readers in various fields without assuming any advanced knowledge in math and physics. I recommend its publication as it is.

---

## Editorial Decision

published